

# Experimental sand burial affects seedling survivorship, morphological traits and biomass allocation of *Ulmus pumila var. sabulosa* in Horqin Sandy Land

Jiao Tang[1,2,], Carlos Alberto Busso[3], Deming Jiang[1*], Ala Musa[1], Dafu Wu[4], Yongcui Wang[1],Chunping Miao[1,2]

[1]Institute of Applied Ecology, Chinese Academy of Sciences, Shenyang, 110016, China;

[2] University of Chinese Academy of Sciences, Beijing, 100048, China;

[3]Departamento de Agronomá-CERZOS (CONICET: Consejo Nacional de Investigaciones Cientficas y Tecnológicas de la República Argentina), Universidad Nacional del Sur, San Andrés 800, 8000 Bahá Blanca, Argentina;

[4]Department of resource and environment, Henan Institute of Science and technology, Xinxiang,453003, China

**Correspondence to:** Deming Jiang ( jiangdeming2016@163.com)

**Abstracts   As** a native tree species, *Ulmus pumila var. sabulosa* (Sandy elm) is widely distributed in Horqin Sandy Land. However, seedlings of this species have to withstand various depths of sand burial after emergence because of increasing soil degradation. So an experiment was conducted to evaluate the changes in the survivorship, morphological traits and biomass allocation buried with different burial depths (unburied, and seedlings buried vertically up to 33, 67, 100 or 133% of the initial mean seedling height). The results showed that partial sand burial treatments (i.e., less than 67% burial) did not influence seedling survivorship, which still reached 100%. However, seedling mortality increased as sand burial was equal to or greater than 100%. Seedling height and stem diameter increased at least by 6 to 14 % with partial burial in comparison with control treatment. Whilst seeding taproot length, total biomass, and relative growth rates at least enhanced by 10%, 15.6%, and 27.6%, respectively, with the partial sand burial treatment. Furthermore, sand burial decreased total leaf area and changed biomass allocation on seedlings, transferring more biomass to aboveground rather than belowground parts. Complete sand burial after seedling emergence inhibited its growth, and even lead to death. Our findings indicated that seedling of sandy elm had a certain resistance to partial sand burial and acclimated to sandy environments. The negative effects of common excessive sand burial after seedling emergence help to understand failures in recruitment of sparse elm



woodland in this study region.

**Keywords**: sand accretion, seedling, sandy elm, morphological, biomass allocation, Horqin sandy land

**Introduction**

Soil genesis is the pivotal process that determines the evolution of soil system and offer services and resources to humankind (Berendse et al., 2015; Niu et al., 2015). Simultaneously, due to increasing population and consumption, external disturbance such as land use intensification has a profound impact on soil genesis process, especially overgrazing and inappropriate cultivation around the world(Brevik et al., 2015; Verheijen et al., 2009; Wang et al., 2016). These excessive human interference changes soil hydrological, geochemical and biological cycles, inducing to serious land degradation primary presented land acidification, salinization and desertification (Bellamy et al., 2005; Foley et al., 2005; Gabarron-Galeote et al., 2013; Smith et al., 2015). Horqin Sandy Land, shaping in the middle period of Pleistocene, was located in the southeast of the Mongolia Plateau, China(Qiu, 1989). Because of climatic changes (rainfall distribution and climate warming and drying) and excessive human activities (settlement, overgrazing and gathering), vegetation degradation and land desertification are becoming more obvious in recent 50 years (Cao et al., 2008; Jiang et al., 2003; Zhang et al., 2004). Combine with the effects of strong winds from March to June, sand moves fast in the horizontal or vertical space leading to different burial depths, which might range from 0.5cm to 56.0cm (Liu et al., 2014).

It is well known that vegetation plays an important role controlling the soil genesis and degradation in the fragile ecosystems (eg. estuarine, desert and sandy land)(Berendse et al., 2015; Cerda, 1998; Miao et al., 2014). And their survival and growth is always restricted by barren and harsh environments. However, compared with plant production and ecological function for

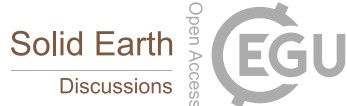

humans, less attention was drawn to the interactions of soil and plants during the long-term evolution and succession
processes under relatively stressful and variable conditions(Foley et al., 2011; Zhao et al., 2004; Zhou et al., 2006). In sandy
ecosystems, instability of the soil surface is one of the most destructive factors and sparse vegetation coverage and loose soil
texture is highly susceptible to sand movements (Liu and Guo, 2005; Yan et al., 2005) Sand movement, the most direct
performance caused by land degradation, is regarded as a selective force to determine colonization, distribution and
establishment of vegetation (Maun, 1994; Maun and Lapierre, 1986). Plants might suffer from varying degrees of sand burial
and evolve different regenerative adaptations in the period of soil seed bank formation, seed germination and following
seedling emergence and development(Li et al., 2014; Qian et al., 2016; Tang et al., 2016). And increasing number of
researches have involved in this field and studies on the effects of sand burial on seedlings have been widely reported on
their survival (Harris and Davy, 1987; Li et al., 2015; Liu et al., 2008; Perumal and Maun, 2006), physiological
characteristics (Shi et al., 2004; Wang et al., 2012; Zhao et al., 2015),and reproduction strategies (Liu et al., 2014; Sun et al.,
2014b; Zhao et al., 2007).In general, there have a threshold sand burial depth for each plant species to maximize its vigor
and the following growth(Maun, 1997). Below the burial level, plant emergence and development have been promoted by
sand burial depth (Qu et al., 2012; Yang et al., 2007);however, exceeded that threshold, a deterioration of seedling vigor and
growth was occur , and even seedling death (Maun, 1997; Maun and Lapierre, 1986).
As a indigenous tree species in this area, *Ulmus pumila var. sabulosa* (sandy elm) was widely distributed in the leeward
slope of fixed and semi-fixed sand dunes and becomes the main proportion of sparse woodlands in Horqin sandy land(Jiang
et al., 2013; Tang et al., 2014; Tang et al., 2013). Since prehistoric times, it is closely relate to human life, which could
provide hardwood for farming tools and furniture, fuel for nomad, fodder because of tender leaves, young fruits and
edible bark(Ma, 1989; Schlütz et al., 2008). At same time, sparse elm woodlands, this landscape type not only offers
shelter for wildlife and domestic animals and paradise for psammophytes, but also prevent soil from wind erosion and
burial in the arid land and semi-arid land, expressing maximum ecological and social function(Yang et al., 2003).
However, even though plenty of seeds germinated without dormancy and seedlings emerged after dispersal in the late spring,
few of seedlings survived were detected in the following field vegetation surveys in the degraded sparse woodlands. This
phenomenon has negative effects on community future structure, and on hampering its recruitment on woodlands
ecosystems.
Despite realization that the effects of sand burial, much of our comprehension and recognition come from ocular
observations at the field rather than from controllable experiments (Maun 1997). Previous studies focused on plant
physiological characteristics and population adaptive strategies to burial in coastal marshes and wetlands (Belcher, 1977;
Cheplick and Grandstaff, 1997; Maun and Lapierre, 1986; Shi et al., 2004; Sun et al., 2010; Sun et al., 2014a). Few has
involved in the effects of sand burial on the establishment and growth of seedlings after their emergence, especially in sandy
environments. To date, no research has been conducted on the effects of continual sand accumulation on sandy elm seedlings
after emergence. This might partially due to the limited, sparse elm woodland area. So we investigated the effects of
experimental sand burial on seedling survivor and growth of sandy elm and provide a theoretical basis for recruitment and
vegetation reconstruction on sandy elm woodlands.

## Materials and methods

### *Study site*

The experiment was conducted at the U'lanaodu Desertification Experimental Station of the Institute of Applied Ecology,

Chinese Academy of Sciences (43°02′N,119°39′E, 480m a.s.l), where located in western Horqin Sandy Land, China. This

site belongs to temperate continental climate. Mean annual temperature and precipitation are 7.3℃ and 315mm, respectively.

Almost 75% of precipitation occurs during June to September in the growing season (Figure 1). Annual average wind speed

is 4.4m/s; the windy season is from March to June (Liu et al., 2012; Miao et al., 2014).The landscape is characterized by

sparse woodlands, sand dunes and lowland areas. Dominant soils are sandy, and major plant species include some shrubs

(e.g., *Salix gordejevii* and *Caragana microphylla L.*),and perennial and annual herbs (e.g., *BassiadasyphyllaL., Agriophyllum*

*squarrosum L) (Cao et al., 2011)*.

### *Experimental methods*

In mid-May 2015, sandy elm seeds were first collected from multiple, mature individuals and then mixed altogether. After

careful selection, uniform and intact seeds were chosen and sowed in plastic pots (45cm diameter, 30cm height). Sandy soil

was taken from nearby woodlands, and it was sieved to remove debris and branches. All seeds were covered by sand to a

depth of 0.5-1.0cm. In a parallel study, we found that that depth was the most suitable for having the greatest percentage and

speed of seedling emergence for sandy elm(Tang et al., 2016). Holes at the bottom of the pots were covered with nylon mesh

to prevent soil loss, while allowing drainage of water at the same time. All pots were watered every three days to keep the

soil moist. Twenty days after sowing, 8 to 12 seedlings emerged; similar seedlings were left in each pot, and the rest was



removed. Mean seedling height (5.4±0.5 cm) was obtained after measuring the height of each seedling in every pot.
Afterwards, seedlings were experimentally buried to either 0(T0, no burial, control treatment) or 33% (T33; 1.8cm), 67%
(T67; 3.6cm), 100% (T100; 5.4cm) or 133% (T133; 7.2cm) soil depth of the original mean seedling height. With this
purpose, sandy soil was added to the pots according to the different burial depths. Each seedling was kept vertical while
buried. Six replicates were used per treatment, so there were a total of 30 pots in this experiment. Meanwhile, 15 randomly
selected seedlings were harvested to determine the original measurements for growth analysis before sand burial.
Surviving seedlings were counted after treatment started of 45 days. They were considered alive in case that    fresh phloem
occurred in both stem and roots, and green tissue in leaf blades. Immediately after burial, seedling height was measured from
the new soil surface level to the seedling apex. Stem diameter was measured close to the burial surface using a vernier
caliper. Meantime, 15 randomly selected seedlings were dug out in each treatment; roots were picked up as intact as possible
from the sandy soil. Taproot lengths were measured, and total leaf area was obtained using a Portable Area Meter
(Li-Cor3000A, Lincoln, Nebraska, USA). Finally, plant organs (leaves, stems and roots) were dried at 80$^{\circ}$C and weighed
after reaching a constant mass for each seedling in the laboratory.
***Calculations***
The (1) relative height growth rate (RHGR, mm.cm$^{-1}$d$^{-1}$), and (2) relative growth rate of seedlings (RGR, mg.mg$^{-1}$.d$^{-1}$) were
calculated according to the following equations (Walck et al., 1999; Zhao et al., 2007):
$$\text{RHGR} = \frac{H_2 - H_1}{H_1(T_2 - T_1)} \qquad (1);$$
$$\text{RGR} = \frac{\ln M_2 - \ln M_1}{T_2 - T_1} \qquad (2);$$



where $H_2$ and $H_1$ were seedling heights at the end and initiation (i.e., immediately before sand burial) of the experiment,
respectively; $M_2$ or $M_1$ were the total dry biomass of seedlings either after 45 days from study initiation or just before sand
burial, respectively; ln was the natural logarithm, and $T_2$-$T_1$was time from sand burial (i.e., 45 days).
*Statistical analysis*
All data were tested for normality and homogeneity of variance prior to analysis. Data were log-transformed if necessary
(Sokal and Rohlf 1995). The effects of experimental sand burial on seedling height, RHGR, plant stem diameter, total leaf
area, RGR, dry biomass and percentage biomass allocation were evaluated by one-way ANOVA. Whenever F tests were
significant, Tukey's test was used to compare treatment means at P<0.05. All statistical analysis used SPSS 21.0 (SPSS Inc.,
Chicago, USA), and drawing was made using Origin Pro 9.0 (Origin Lab Corp, USA).

**Results**
*Effects of continual sand burial on seedling survival*
The effect of sand burial depth on seedling survival was significant ($F_{4,235}$=38.339, P=0.000). During the whole study,
seedling survival was 100% on the unburied (T0) and partial burial treatments (T33, T67). Simultaneously, seedling survival
(84.48±8%) was significantly lower in the completely sand burial treatment (T100). No seedling survived after burial depth
exceeded the mean original height of seedlings (i.e., on T133).
*Changes of morphological seedling traits in response to sand burial*
*Seedling height and height growth rate*



Seedling height was significantly affected by sand burial depths after 45 days of burial ($F_{3,56}$=139.978, P=0.000).The highest
seedling height of 22.35cm was observed in the T33 treatment, which was significantly greater than that in the T67 treatment
(Figure 1A; P<0.05). Height of seedlings in the control treatment of 16.28cm was significantly lower than that in the T33 and
T67 treatments, but higher than in the T100 treatment of 10.66cm (Figure 1A; P<0.05).
The relative seedling height growth rate (RHGR) was significantly affected ($F_{3,56}$= 286.877; P=0.000) by sand burial of 45
days(Figure 1B). Highest of 0.0574mm.cm$^{-1}$d$^{-1}$ and lowest of 0.0232 mm.cm$^{-1}$d$^{-1}$ in relative growth rates for seedling height
were shown in the T33 and T100 treatments, respectively (Figure 1C). The pattern of change with burial depth was similar to
that described for seedling height (Figure 1A); values were greater in the control than 100% covered by sand (Figure 1B;
P<0.05). ***Seedling stem diameter, taproot length and total leaf area***
After 45 days from initial study, stem diameter ($F_{3,56}$=26.669, P=0.000), taproot length ($F_{3,56}$=30.942, P<0.001) and total leaf
area ($F_{3,56}$=35.961, P<0.001) of seedlings were also affected by sand burial (Figure 3 A, B, C). Stem diameter was 20%
greater (P<0.05) in the T33 than in the T0 treatment of 1.793mm (Figure 3A). However, diameters of stems were similar in
the control and T67 treatments (Figure 3A). The values in the T100 treatment, nevertheless, were 13.4% lower than unburied
control of 1.493mm (Figure 3A). While taproot length was lowest in the control treatment of 19.39cm, it was highest in the
T67 treatment (Figure 3B; 35.7% higher than in the control; P<0.05). The total leaf area of seedlings was significantly
greater in the control of 23.87cm$^2$ than in the T67 and T100 treatments (Figure 3C; P<0.05). The lowest total leaf area,
however, was found in the T100 treatment of 16.89cm$^2$ (Figure 3C).
**Effects of sand burial on biomass growth and relative growth rate**



### Dry biomass and percentage allocation

There were significant differences in the total seedling biomass ($F_{3,56}$=129.949, P=0.000) and its component organs after the experiment: leaves ($F_{3,56}$=93.965, P=0.000) and roots ($F_{3,56}$=50.474, P=0.002). The only exception was for seedling stem biomass ($F_{3,56}$=2.017, P=0.122) which was similar in all sand burial treatments (Figure 4A). And seedling organs also showed a similar pattern in their dry biomass. Greatest total biomasses were reached in the T33 and T67 treatments of 369.65mg and 372.50mg, respectively (Figure 4A). While total biomass of seedlings was significantly lower in T100 treatment of 272.67mg than in those treatments (Figure 4 A). Patterns shown for the biomasses of leaves and roots among treatments were similar to those shown for the total biomass of seedlings (Figure 4A).

Significant differences were found in allocation of seedling biomass to leaves ($F_{3,56}$=12.841, P=<0.001), stems ($F_{3,56}$=27.579, P=0.000) and roots ($F_{3,56}$=7.594, P=<0.001). On leaves, percentage biomass allocation of 52.9% and 54.2% was greatest in the T33 and T67 treatments, and lowest in the control and T100 treatments (Figure 3B). Percentage biomass allocation to stems was greatest in the T100 treatment of 24.1% and the value of stems was greater in the control of 20.9% than in the T33 and T67 treatments (Figure 4B). Finally, percentage   biomass allocation to roots was similar in the control, T33 and T67 treatments (Figure 4B); values determined in the T100 treatment for this organ were significantly lower than in the control and T33 treatments (Figure 4B).

### Seeding relative growth rate

The relative growth rate of seedlings was significantly affected by sand burial after experiment ($F_{3,56}$=136.370, P=0.000). Greatest relative growth rate values of 0.031mg.mg$^{-1}$day$^{-1}$ were shown in the T33 and T67 treatments (Figure 5). These



values were significantly greater than those found in the control (Figure 5). Lowest relative rate of growth was determined
on seedlings grown in the T100 treatment, just 0.026 mg mg$^{-1}$day$^{-1}$(Figure 5).

**Discussion**
***Seedling survivorship in response to continual sand burial***
In Sand land regions, seedlings might be buried to different depths after emergence by the end of the windy season from late
spring to early summer(Chen and Maun, 1999). After 45-day-experiment, survivorship remained unchanged when seedlings
were either unburied or exposed to partial [i.e.,1.8 (T33) and 3.6 (T67) cm soil depth] sand burial. These results agreed with
studies by He et al. (2008), Liu et al. (2008) and Qu et al. (2012), which reported that vigor of *Artemisia halodendron,*
*Corispermum macrocarpum* and *Caragana microphylla* was either maintained or increased by moderate sand burial in
Horqin Sandy Land. Survivorship of these shrubs, however, varied among plant species once their seedlings were covered by
sand either equal to or more than 100% of their height. Whilst the values decreased significantly by a mean of 15.6% given
completely covered the whole, original seedling height [i.e., 5.4cm soil depth (T100)]. Maun (1981, 1997) and Disraeli
(1984) indicated that a certain tolerance of sand burial was an effective strategy for determining the survival and subsequent
establishment of seedlings in sandy environments. While sand burial exceeded 33% of the original seedling height [i.e.,
7.2cm soil depth (T133)], there was no seedling survival, only withered and rotted in the soil. Harris and Davy (1987) and
Perumal and Maun (2006) believed that plant energy exhaustion and suppression of photosynthesis most likely severely
reduced seedling survival under extreme shade, sand burial environments. Most seedlings of the grass *Distichlis spicata* died



when completely covered by sand in North America (Brown, 1997; Li et al., 2015). While some *Artemisia.squarrosum*
seedlings remained still alive even though sand burial depths reached 266% of initial seedling height in the Horqin Sandy
Land (Li et al. 2015). Compared with other species, seedlings of sandy elm showed a certain resistance to partial sand burial,
but complete sand burial significantly increased mortality.
*Effects of sand burial on seedling morphological traits*
Sand burial can alter the biological (e.g., the photosynthetic active radiation) and abiotic (e.g., soil temperature, moisture
availability) environment of living plants which influence morphological structures and subsequent growth(Disraeli, 1984;
Sun et al., 2014b).Our results demonstrated that various seedling morphological traits (i.e., height; stem diameter; taproot
length; total leaf area; dry biomass; partitioning of biomass to shoots and roots; RHGR and RGR ) were greater at the
intermediate(i.e., T33 and T67) than at the more extreme (i.e., T100 and T133) sand burial depths. Changes on these traits
were determinant in allowing the obtained seedling developments in the different study burial environments.
In our experiment, seedling height of sandy elm was greater in the partial than in the unburied and completely burial
treatments, which indicated that partial sand burial stimulated stem elongation. This acceleration could be explained by the
fact that the processes of growth and elongation benefited water and nutrient uptake in arid and semi-arid regions (Li et al.,
2015). This observation is also consistent with previous research reported by Disraeli (1984) that partial burial stimulated
growth of *Ammophila Breviligulata* in coastal dunes of northeastern North America (Disraeli, 1984) and Belcher (1977) that
seedling heights of *Rosa rugosa* were higher in the partial than in the unburied and partial continuous sand burial treatments
in desert.

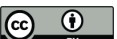



When seedlings re-emerged from the new experimental soil surface after the various sand burial treatments, it was
critical that the ability to compensate from the effects of sand burial. Seedling height growth rate was a critical parameter to
determine the speed of growth. The greater RHGR of seedlings in the partial (T33 and T67) than in the unburied and
completely burying treatments (T100) was an indication that partial burial did contribute to seedling height after a
45-day-growth period via accelerating the speed of growth in height. Nevertheless, the greatest seedling growth in height in
the T33 treatment came from its greatest RHGR than any other treatments. The same phenomena have been observed by Liu
et al. (2008) and Miao et al. (2012) who also found that shallow burial depths could promote the relative growth rates in
height of *Salix gordejevii*, *Artemisia wudanica* and *A. halodendron*. However, some species(e.g., *A.gmelinii*) have decreased
their growth rates as a result of sand burial (Liu et al., 2008). These findings confirmed that growth resistance to sand burial
might be species-specific.
Compared with the unburied treatment, partial burial treatments fostered increments in stem diameter and taproot length. Sun
et al. (2014a) also showed that seedling diameter and taproot length of *Suaeda salsa* were increased by a partial burial
treatment in the coastal marches of the Yellow River estuary, China. Caldwell et al. (1998) found that increases in taproot
length were conducive to a greater nutrient and water uptake from deeper soil depths. Even more, water and nutrients
obtained from deep depths by dicots species could be released at shallower soil depths if soil is drier than plant tissues at
those depths (Caldwell et al. 1998). If this happens, both water and nutrients can be parasitized by shallower-rooted species,
with subsequent implications in nutrient cycling (Cardon et al. 2013). Total leaf area, however, was either similar or lower in
the partial burial than in the unburied treatment. This was in agreement with the results obtained by Liu and Guo (2005) who



noted that increasing depth of sand burial decreased the total leaf area of *Caragana intermedia*.

236       An appropriate resource allocation is essential for plant establishment and growth (Bazzaz FA, 1997). Also, plants shift

resource allocation to minimize external changes (Maun, 1997; Ni et al., 2015). Numerous findings, especially those living
in sandy environments, reported that plants could withstand various depths of sand burial by changing biomass allocation.
Some species (e.g. *Artemisia ordosia, Elymusfarctus*) would transfer biomass from underground to leaf and stem organs
(Brown, 1997; Li et al., 2010), while others (e.g.,*Caraganamicrophylla*, *Nitrariasphaerocarpa*) either maintained or
increased biomass allocation to roots(He et al., 2008; Sykes and Wilson, 1990). In our experiment, it was somewhat
surprising that no difference in the dry biomass of seedling stems was observed among all sand burial treatments even
though seedlings showed an increased stem diameter in the T33 sand burial treatment, that was probably due to
different-diameter, fresh stems of this species showed different water contents(Zhao et al., 2015).
Partial burial treatments determined a greater dry biomass for leaves, roots and the whole seedlings than values shown for
these traits in the unburied and completely sand burial treatments. These results were expected as previous studies showed
that 67% burial of seedlings of the shrub *Caragana intermedia* determined a greater biomass allocation to leaves and stems,
and a lower one to roots, compared with values in the unburied control(Xu et al., 2013). Meanwhile, there was a trend for an
increasing aboveground (leaf + stem), and a decreasing belowground allocation with increasing burial depth. Nearly 50% of
the total seedling biomass corresponded to leaves, which suggested that leaves were important for sandy elm, which
sufficient leaves could guarantee normal photosynthesis and maintain evapo-transpiration for sandy elm exposed to high
temperature environments in the growing season(Dulamsuren et al., 2009).



Relative growth rates measure the mean efficiency rate of producing new biomass (Walck et al., 1999). Dalling and Hubbell
(2002) found that seedling growth rate instead of biomass was the key to determine the successful seedling establishment. In
our experiment, relative growth rates were higher in the partial than in the other treatments, indicating that moderate sand
burial was beneficial for rapid growth after seedling sand burial. However, all sandy elm seedling mass relative growth rates
were small compared with those of other plant species (e.g.,*Artemisia wudanica*,*Solidagoshortii* and *Solidago. altissima*) in
the same area. This suggests that sandy elm biomass accumulation was slow during the first growing season (Brown, 1997;
Liu et al., 2014).
Biomass accumulation and allocation, and the plasticity of various morphological traits, were critical in determining the
response, and survivorship and development of sandy elm seedlings in sandy environments. However, this burial study was
conducted in a common garden. Under natural, rangeland conditions, some factors (e.g. abrasion of plant tissues by sand
grains; gnaw by herbivores and granivores) might reduce or eliminate some of the positive effects of the partial burial
treatments (Baker et al., 2009; Dulamsuren et al., 2009). Therefore, more comprehensive studies on physiological and
biochemical mechanisms which are involved in sandy elm seedling survivorship and performance under sand burial
conditions are necessary in future research.
**Conclusion**
Sand burial affected seedling survivorship, morphological traits, and biomass production and allocation of *Ulmus pumila var.*
*sabulosa*. Seedlings of sandy elm showed a certain resistance to sand burial. Partial sand burial treatment did not influence
seedling survivorship, but complete sand burial significantly increased mortality. Compared with the unburied treatment,

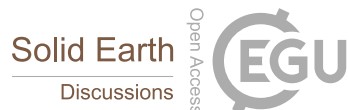

seedling height, absolute and relative height growth rates, taproot length, total biomass and relative growth rates were
stimulated because of partial sand burial. At the same time, percentage biomass allocation of seedlings was changed; these
shifted more biomass to aboveground organs to sustain photosynthesis and evapo-transpiration in response to the sand burial
depth. Complete sand burial after seedling emergence, however, could inhibit their growth, and even result in seedling death;
this is why this burial depth should be controlled by making enclosures and increasing vegetation coverage. The observed
variation in all parameters indicated that *Ulmus pumila var. sabulosa* could tolerate partial sand burial, and that it could
acclimate to sandy environments.

**Acknowledgements**
This study was supported by National Natural Science Foundation of China (31370706) and the U'lanaodu Desertification
Experimental Station of the Institute of Applied Ecology, Chinese Academy of Sciences. We thank Yongming Luo ,Xuehua
Li, Quanlai Zhou, Hongmei Wang, Meiyu Jia, Ya Liu and Xu Han for assistance during the experiment. Thanks Authony
Davy from University of East Anglia for guiding of statistical analysis in the experiment.

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



## Figure legends

**Fig. 1** Dynamics in precipitation and temperature (T) from 1980 to 2014 in U'lanaodu. The solid line shows the average annual precipitation and the dashed line the average precipitation from June to September, during that period. The insert shows the average monthly precipitation (mm) and temperature (℃) during 1980 to 2014.

**Fig.2** Seedling height and relative growth rate for height (RHGR) of *Ulmus pumila var.sabulosa* exposed to various sand burial treatments during a 45-day- growth period. These treatments included sand burial of seedlings to a depth equivalent to 33 (T33), 67 (T67) or 100% (T100) of the mean seedling height at the initiation of the study (see the Material and Methods Section for further details). Each histogram is the mean ±1 S.E. of n=15.

**Fig.3.** Stem diameter, taproot length and total leaf area on seedlings of *Ulmus pumila var.sabulosa* exposed to various sand burial treatments during a 45-day-growth period. These treatments included sand burial of seedlings to a depth equivalent to 33 (T33), 67 (T67) or 100% (T100) of the mean seedling height at the initiation of the study. Each histogram is the mean ±1 S.E. of n=15.

**Fig.4** Seedling dry biomass and percentage biomass allocation to leaves, stems and roots on seedlings of *Ulmus pumila var.sabulosa* exposed to various sand burial treatments during a 45-day-growth period. These treatments included sand burial of seedlings to a depth equivalent to 33 (T33), 67 (T67) or 100% (T100) of the mean seedling height at the initiation of the study. Each histogram is the mean ±1 S.E. of n=15.



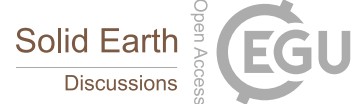

**Fig.5** Relative growth rates on seedlings of *Ulmus pumila var.sabulosa* exposed to various sand burial treatments during a
45-day-growth period. These treatments included sand burial of seedlings to a depth equivalent to 33 (T33), 67 (T67) or 100%
(T100) of the mean seedling height at the initiation of the study. Each histogram is the mean $\pm 1$ S.E. of n=15.




















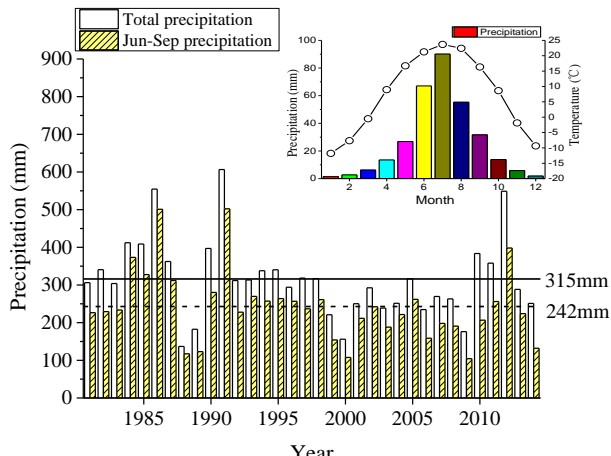


Figure1

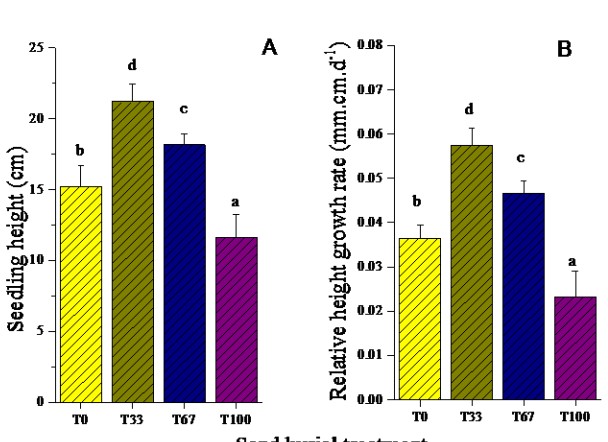


Figure2





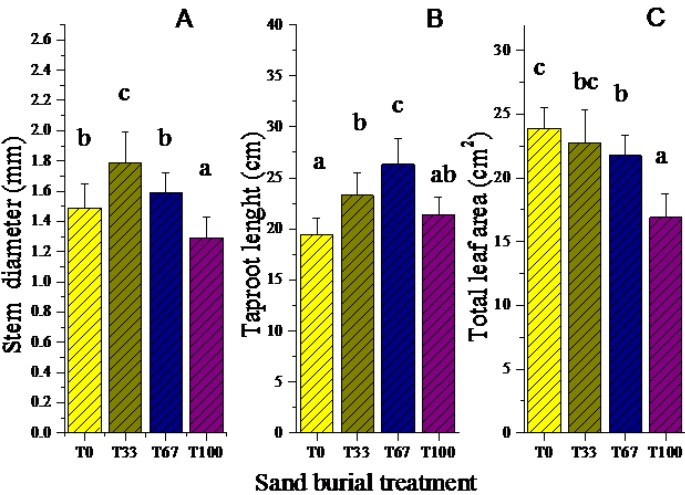


492                                                                Figure3

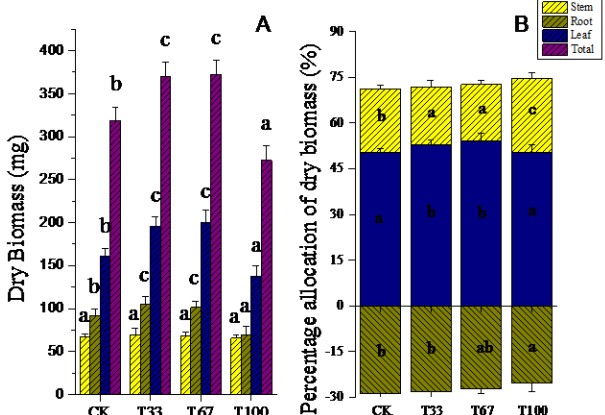

494                                                                Figure4



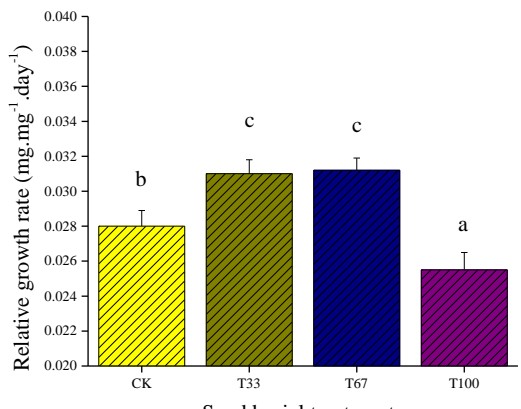



497                                        Figure5
