# Peer review of "Experimental sand burial affects seedling survivorship, morphological traits and biomass allocation of *Ulmus pumila var. sabulosa* in Horqin Sandy Land"

_Solid Earth, 2016_

## Referee Comment (RC1) · Anonymous Referee #1 · 5 Apr 2016

This paper aims to study the effects of sand burial effects on seedling survivorship, morphological traits and biomass allocation. The topic is good, but there are only effects of sand burial on seedling survivorship, morphological traits and biomass allocation. The burial effects on seedling survivorship should may be results from burial effects on seedling morphological and biomass allocation strategies. So, I think that the authors should added the contents of burial effects on seedling survivorship based on seedling morphological and biomass allocation strategies. Secondly, the burial effects on seed germination and seedling emergence may be focus on the effects of irridiance and temperature effects on seed germination and seedling emergence, so some dicsuccion

contents on this aspects should be addedin in the discussion section. Some references can be read, Seedling performance within eight different seed-size alpine forbs under experimental irradiance and nutrient gradients; Germination strategies of twenty alpine species with varying seed mass and light availability; Plant seedling performance traits impact on successful recruitment in various microhabitats for five alpine Saussurea species;SEEDLING RECRUITMENT OF FORB SPECIES UNDER EXPERIMENTAL MICROHABITATS IN ALPINE GRASSLAND, etc. And, there are some format errors in the text and in the references, suggest the author should avoid the appearance of these errors in all the text.

---

## Referee Comment (RC2) · Anonymous Referee #2 · 6 Apr 2016

General comments: Ulmus pumila v. sabulosa is an indigenous tree species in Horqin Sandy Land, which is main proportion of sparse woodlands in this area. How to facilitate it to naturally regenerate is an important and interesting problem in ecological restoration of degraded grassland. The manuscript investigates the survivorship, morphological traits and biomass allocation of U. pumila v. sabulosa buried with different burial depths. The topic is interesting. The results will be useful, in that it gives some proofs and facilitates further understanding for plant seedling survival strategy in eroded sandy environment. The experimental set-up and the study methods used seem adequate. The paper is within the scope of Solid Earth, and the research was

performed in an interesting area of China. I recommend that the paper is reconsidered for its publication in Solid Earth after some corrections are done.

Special comments: 1 In introduction part, the scientific questions of this study was not clear, and scientific hypothesis and scientific significance were not clearly mentioned. I suggest authors revise this section. 2 In Experimental methods, "seedlings were experimentally buried to either 0, 33%, 67%, 100%, or 133% soil depth of the original mean seedling height." I wonder why 133% of seedling height treatment was conducted. 100% buried means no photosynthesis occurred, and will result in the death of the seedling. I think the treatment of 133% is unnecessary. 3 In Statistical analysis, using the methods of one-way ANOVA and multiple comparisons to analyze were not adequate, in that the experiment was a gradient test (0, 33%, 67%, 100%), I suggest you try to use regression analysis to analyze the relationship between variables and buried ratios. 4 Fig.1 was useless.

---

## Referee Comment (RC3) · Anonymous Referee #3 · 6 Apr 2016

I recommend review this work again after some improvements. I appreciate the authors clarify my doubts.

L31 "transferring more biomass to aboveground rather than belowground parts." It is interesting, but not be well supported. I suggested to compare the above ground biomass (i.e. stem+leaf) and below ground biomass (i.e. root) respectively. The root biomass was reported in the previous version, but the mean and s.d. values were not shown in the results part.

L109 Before the start of burial treatment experiment, Is the number of seedlings in

each pot same with each other? What is the number of seedlings in each pot?

L116 "Immediately after burial. . ." Does it mean after the end of burial treatment experiment?

L139 The second degree of freedom in F test could not be 235 in the experiment described in current status. Please verify it!

L147 "Height of seedlings in the control treatment of 16.28cm was significantly lower than that in the T33". The mean value is not enough. The s.d value should be reported.

L149 The mean and s.d. values of height growth rate should be reported. L155 The mean and s.d. values of stem diameter should be reported. If it is nessesary, the mean and s.d. values of other indices should be reported. It is helpful for evaluating the effects of sand burial on the growth of elm seedlings.

Technical corrections L22 "as" should not be bold

L37 "Horqin sandy land" should be "Horqin Sandy Land"

L58 A full stop is needed before "Sand movement"

L71 "Horqin sandy land" should be "Horqin Sandy Land"

L82 A comma is needed between "Maun" and "1997".

L94 "where" should be "which"

L97 It should be 4.4 m s-1ïijĹsuperscriptïijĽ

L100 "Cao et al., 2011" should not be in italics

L115 A blank space between "that" and "fresh" is unnecessary

L 123 mm.cm-1d-1 should be mm•cm-1•d-1. Same mistakes in other place should be verified.

L139 P =0.000 should be P<0.001. The same problem exists in other place, please

verify it.

L153 "Seedling stem diameter, taproot length and total leaf area" should move to the next line

L175 A blank space between "percentage" and "biomass" is unnecessary

L186 "Sand" should be sand

---

## Referee Comment (RC4) · Anonymous Referee #4 · 18 Apr 2016

GENERAL COMMENTS:

The work of Tang et al. compares the effects of partial and total sand burial on sandy elm seedlings: mortality, morphological traits, root allocation, etc. The study has been carried out in Horqin Sandy Land (China) where problems with wind erosion are a common feature. Results show an interesting reflection: Sandy elm seedlings are sufficiently resistant to partial burial. However actions to reduce wind erosion in this part of China are a key issue because total burial effects are largely negative.

This research could be valuable mainly by two reasons: originality and usefulness.

However, in my opinion, this article is not yet worked enough to merit publication in a prestigious journal such as SOLID EARTH, at least this version. English should be checked by a native speaker. Introduction should be restructured and rewritten, and too many mistakes can be still observed in the text (please see specific comments below). Therefore, I am going to suggest to the Editor a Major Revision in order to be able to give more specific suggestions in the next version.

SPECIFIC AND TECHNICAL COMMENTS:

Title

"(China)" should be mentioned in the title.

Filiations

"Department of Resource and Environment, Henan Institute of Science and Technology" Please use capital letters always that it is possible.

Abstract

Page 1, line 22: "Abstracts" Please delete the final "s" and replace in a single row.

Page 1, line 22: "Horqin Sandy Land (China)" SE is a worldwide journal.

Page 1, line 24: "soil degradation" What is the dominant process? Wind erosion? Please be more precise. An abstract aims to simplify the reading comprehension.

Page 1, line 25: "allocation buried at different depth: unburied and seedlings buried..." Please correct it.

Keywords

Please delete "sand accretion"

Introduction

This chapter is a little disorganized, i.e. it is not easy to read. Please restructure it.

Soil genesis and the role of vegetation should be written in the same paragraph.

Page 2, line 43: "cultivation around the world (Brevik...)". Please separate it.

Page 2, lines 43 and 44: "This excessive human interference changes" Please correct it

Page 2, line 45: "Gabarrón-Galeote" Please use accent marks in Latin surnames.

Page 2, lines 46-51: "These sentences should be written more forward in the Introduction section". It does not make any sense speaking about soil genesis, Horqin Sandy Land and the role of vegetation. Please be consistent.

Page 2, line 46: "Horqin Sandy Land is located" Please correct it.

Page 2, line 47: "China (Qiu, 1989)" Please separate it (throughout the text, it is a common mistake).

Page 2, lines 52 and 53: "vegetation plays an important role controlling soil genesis and degradation in fragile ecosystems such as estuarine, desert or sandy land" Please correct it.

Page 2, line 53: "Cerdà" Please use accent marks in Latin surnames.

Pages 2 and 3, lines 54-56: "Please remove these lines. They are superfluous"

Page 3, line 57: "Here it should begin the second paragraph of the Introduction section".

Page 3, line 57: "In sandy ecosystems, instability of the soil surface is one of the most destructive factors. Furthermore, sparse vegetation coverage and loose soil texture are highly susceptible..." Please correct it.

Page 3, line 62: "development (Li..." Please separate it.

Page 3, line 62: "An increasing" Please correct it. That should be the beginning of a third paragraph.

Page 3, line 65: "), and reproduction" Please separate it.

Page 3, line 67: "following growth (Maun, 1997)" Please separate it.

Page 3, line 68: "; however" Please separate it.

Page 3, line 71: Idem

Page 4, lines 74 and 76: Idem

Material and methods

Study site

Page 5, line 94: "480 m a.s.l.) located in..." Please correct it.

Number and units should be separated.

Places do not belong to any climate. They are characterized or dominated by some types of climate. Please correct it.

Experimental methods

Page 5, line 102: "from multiple mature individuals" Please correct it.

Page 5, line 105: "0.5 – 1.0 cm" Please separate number and units.

Page 5, line 106: "sandy elm (Tang et al., 2016)" Please separate text and citation

Page 6, line 110: Idem

Calculations

What does it mean each unit: e.g. mm.cm-1d-1?

Results

In my opinion, there are too many sub-chapters. Furthermore, sometimes figures are not mentioned or bad mentioned in the text (e.g. figure 1, page 8) and some tables

could be useful for readers.

Number and units, text and parenthesis, etc. should be separated (throughout the text).

Page 8, line 153: The title of a sub-chapter should be presented in a single row

Page 8, line 156: Why do you use cm in some cases and mm in other cases?

Discussion

It should be checked by a native speaker. There are too many grammar and style mistakes.

References

They are presented in a different style: font, interlines, etc.
* * *

---

## Author Comment (AC1) · 26 May 2016

RC1: The topic is good, but there are only effects of sand burial on seedling survivorship, morphological traits and biomass allocation. The burial effects on seedling survivorship may be results from burial effects on seedling morphological and biomass allocation strategies. So, I think that the authors should add the contents of burial effects on seedling survivorship based on seedling morphological and biomass allocation strategies. AC1: Thanks for your significant suggestions. And we have added some contents of burial effected on seedling survivorship based on seedling morphological and biomass allocation strategies in the discussion of the revised manuscript.

RC2: The burial effects on seed germination and seedling emergence may be focus on the effects of irradiance and temperature effects on seed germination and seedling emergence, so some discussion contents on these aspects should be added in the discussion section. Some references can be read, Seedling performance within eight different seed-size alpine forbs under experimental irradiance and nutrient gradients; Germination strategies of twenty alpine species with varying seed mass and light availability; Plant seedling performance traits impact on successful recruitment in various microhabitats for five alpine Saussurea species; Seedling recruitment of forbs species under experimental microhabitats in alpine grassland, etc. AC2: We looked up some relevant references and found that sandy elm had a higher transpiration rate and stomatal conductance with lower photosynthesis water-use efficiency and less sensitivity to high temperature and irradiance, compared with other native tree species such as Malus baccata, Prunus padus and Pinus sylvestris, especially in the midday. We have added it to the discussion section. The reviewer's recommended references are of importance for improving our manuscript. We have read carefully and several relevant sentences were cited in our revised manuscript.

RC3: There are some format errors in the text and in the references, suggest the author should avoid the appearance of these errors in all the text. AC3: We have checked and corrected the format in the text and in the references.

Thanks for your valuable advices for our manuscript.

Please also note the supplement to this comment:
http://www.solid-earth-discuss.net/se-2016-55/se-2016-55-AC1-supplement.pdf

**Supplement:**

**Experimental sand burial affects seedling survivorship, morphological traits and biomass allocation of *Ulmus pumila* var. *sabolusa* in Horqin Sandy Land, China**

Jiao Tang[1,2], Carlos Alberto Busso[3], Deming Jiang[1*], Ala Musa[1], Dafu Wu[4], Yongcui Wang[1],Chunping Miao[1,2]

[1]Institute of Applied Ecology, Chinese Academy of Sciences, Shenyang, 110016, China;

[2] University of Chinese Academy of Sciences, Beijing, 100048, China;

[3]Departamento de Agronomía-CERZOS (CONICET: Consejo Nacional de Investigaciones Científicas y Tecnológicas de la República Argentina), Universidad Nacional del Sur, San Andrés 800, 8000 Bahía Blanca, Argentina;

[4]Department of Resource and Environment, Henan Institute of Science and Technology, Xinxiang,453003, China

**Correspondence to:** Deming Jiang ( jiangdeming2016@163.com)

**Abstract.** As a native tree species, *Ulmus pumila* var. *sabolusa* (sandy elm) is widely distributed in Horqin Sandy Land, China. However, seedlings of this species have to withstand various depths of sand burial after emergence because of increasing soil degradation, which is mainly caused by overgrazing and climate change. An experiment was conducted to evaluate the changes in its survivorship, morphological traits and biomass allocation when seedlings buried at different burial depths : unburied, controls and seedlings buried vertically up to 33, 67, 100 or 133% of their initial mean seedling height. The results showed that partial sand burial treatments (i.e., less than 67% burial) did not reduce seedling survivorship, which still reached 100%. However, seedling mortality increased when sand burial was equal to or greater than 100% In comparison with the control treatment, seedling height and stem diameter increased at least by 6 and 14% with partial burial, respectively. In the meantime, seeding taproot length, total biomass, and relative mass growth rates at least enhanced by 10%, 15.6%, and 27.6%, respectively, with the partial sand burial treatment. Furthermore, sand burial decreased total leaf area and changed biomass allocation in seedlings, partitioning more biomass to aboveground organs (eg., leaves and stems) and less to belowground parts (roots). Complete sand burial after seedling emergence inhibited its re-emergence and growth, even leading to its death. Our findings indicated that seedlings of sandy elm had some resistance to partial sand burial and were adapted to sandy environments from an evolutionary perspective. The negative effects of excessive sand burial after seedling
emergence might help to understand failures in recruitment of sparse elm in the study region.

**Keywords**: sand burial, seedling, sandy elm, morphological, biomass allocation, Horqin Sandy Land

**Introduction**

Horqin Sandy Land, shaped in the middle the Pleistocene period, is located in the southeast of the Mongolia Plateau, China (Qiu, 1989). Because of climatic changes (rainfall distribution and global warming) and excessive human disturbances (i.e., over-utilization of renewable natural resources), vegetation degradation and land desertification have become more obvious in the past 50 years (Cao et al., 2008; Jiang et al., 2003; Zhang et al., 2004). Sand moves fast in the horizontal or vertical space because of the effects of strong winds during spring and summer, leading to different burial depths, which might range from 0.5cm to 56.0cm (Liu et al., 2014).

Soil genesis is the pivotal process that determines the evolution of the soil system, and offers services and resources to mankind (Berendse et al., 2015; Niu et al., 2015). Simultaneously, disturbances (such as land-use intensification and overgrazing) have a profound impact on the soil genesis process because of the increasing population and consumption (Brevik et al., 2015; Verheijen et al., 2009; Wang et al., 2016). Excessive human interferences change soil hydrological, geochemical and biological cycles, inducing serious land degradation such as acidification, salinization and desertification (Bellamy et al., 2005; Foley et al., 2005; Gabarrón-Galeote et al., 2013; Smith et al., 2015). It is well known that vegetation plays an important role in controlling soil genesis and degradation in fragile ecosystems such as estuarine, desert and sandy lands (Berendse et al., 2015; Cerdà, 1998; Miao et al., 2014). Moreover, vegetation functional traits (e.g., morphology and establishment) in response to environmental stress (e.g., nutrient deficiency, water deficit, high irradiance, extreme


temperatures) might be important adaptive life history adaptive strategy (Miner et al., 2005; Wang et al., 2014). Plants usually have to face a trade-off between survival and growth in response to environmental changes by regulating their phenotypic plasticity (e.g., biomass allocation, relative growth rate) and/or physiological traits (e.g., antioxidant enzyme activities, membrane permeability, contents of osmotic substance) (Li et al., 2015; Qu et al., 2012; Tian et al., 2015; Wu et al., 2013).

In sandy ecosystems, instability of the soil surface is one of the most damaging factors to biological activity. Furthermore, a sparse vegetation cover, and a loose soil texture are highly susceptible to sand movements (Liu and Guo, 2005; Yan et al., 2005). Sand movement, the most direct evidence caused by land degradation, is regarded as a selective force, determining colonization, establishment and establishment of vegetation (Maun, 1994; Maun and Lapierre, 1986). Plants might respond differently to various degrees of sand burial, and evolve different regenerative adaptations during the periods of soil seed bank formation, seed germination, and seedling emergence and development (Li et al., 2014; Qian et al., 2015; Tang et al., 2016).

As an indigenous tree species, *Ulmus pumila* var. *sabolusa* (sandy elm) has been widely distributed in the leeward slope of fixed and semi-fixed sand dunes and became the main component of sparse woodlands of the Horqin Sandy Land (Jiang et al., 2013; Tang et al., 2014; Tang et al., 2013). Since prehistoric times, it has been closely relate to human life, providing hardwood for farming tools and furniture, fuel for nomad, and fodder from its tender leaves, young fruits and edible bark (Ma, 1989; Schlütz et al., 2008). In addition, the sparse-elm woodlands, not only offered shelter for wildlife and domestic animals and a suitable environment for psammophytes; but also protected soil from wind erosion and burial,

Studies on the effects of sand burial have been widely reported in the field of seedlings survival (Belcher, 1977; Cheplick and Grandstaff, 1997; Harris and Davy, 1987; Li et al., 2015; Liu et al., 2008; Perumal and Maun, 2006), physiological characteristics (Shi et al., 2004; Wang et al., 2012; Zhao et al., 2015), and reproductive strategies (Liu et al., 2014; Sun et al.,

2014; Zhao et al., 2007) of coastal marshes and wetlands plants. In general, it appears to be a threshold sand burial depth for each plant species to matain vigor and the subsequent sustain growth (Maun, 1997). Below that burial level, plant emergence and development have been promoted by increasing sand burial depth (Qu et al., 2012; Yang et al., 2007). Once the threshold have been exceeded, a deterioration of seedling vigor and reduced growth has occur, and even leaded to seedling death (Maun, 1997; Maun and Lapierre, 1986). However, no research has been conducted on the effects of continual sand accumulation on sandy elm seedlings after emergence to date, because of the limited area of sparse elm woodland. So we investigated the effects of experimental sand burial on seedling survivorship and growth of sandy elm. The main objectives of this study were: (1) to evaluate the effect of sand burial on seedling survivorship; (2) to access the changes on seedling morphological traits and biomass allocation in response to sand burial, and (3) to explain the failure of sandy elm regeneration, and provide a theoretical basis for achieving a successful recruitment and vegetation establishment on sandy elm woodlands.

**Materials and methods**

*Study site*

The experiment was conducted at the Wulanaodu Desertification Experimental Station of the Institute of Applied Ecology,

Chinese Academy of Sciences (43 °02′N,119 °39′E, 480 m a.s.l) located in western Horqin Sandy Land, China (Figure 1).

This site experiences temperate continental climate. Mean annual temperature and precipitation are 7.3℃ and 315 mm, respectively. Almost 75% of precipitation occurs from June to September during the growing season. Annual average wind speed is 4.4 m s$^{-1}$; the windy season is from March to June (Liu et al., 2012; Miao et al., 2014). The landscape is characterized by sparse woodlands, sand dunes and lowland areas. The dominant soils are aerolian soils, and major plant species include some shrubs (e.g., *Salix gordejevii* and *Caragana microphylla*), and perennial and annual herbs (e.g.,

*Bassiadasyphylla*, *Agriophyllum squarrosum*) (Cao et al., 2011) .

*Experimental methods*

In mid-May 2015, sandy elm seeds were first collected from multiple, mature individuals and then mixed altogether. After careful selection, uniform and intact seeds were chosen and sowed in plastic pots (45 cm diameter, 30 cm height). Sandy soil was taken from nearby woodlands, and it was sieved to remove debris and branches. All seeds were covered by sand to a

depth of 0.5-1.0 cm. In a parallel study, we found that that depth was the most suitable promoting the greatest percentage and speed of seedling emergence for sandy elm (Tang et al., 2016). Holes in the bottom of the pots were covered with nylon mesh to prevent soil loss, while allowing drainage of water. All pots were watered every three days to keep the soil moist. Twenty days after sowing, 8 to 12 seedlings emerged; eight similar seedlings were retained in each pot, and the rest were removed. Mean seedling height (5.4±0.5 cm) was obtained after measuring the height of each seedling in every pot. Afterwards, seedlings were experimentally buried to either 0 (T0, no burial, control treatment) or 33% (T33; 1.8 cm), 67% (T67; 3.6 cm), 100% (T100; 5.4 cm) or 133% (T133; 7.2 cm) soil depth of the original, overall mean seedling height. For this purpose, sandy soil was added to the pots according to the different burial depths. Each seedling was kept vertical while buried. Six replicates were used per treatment, so there was a total of 30 pots in this experiment. Meanwhile, 15 randomly selected seedlings were harvested to determine the original measurements for growth analysis before sand burial.

Surviving seedlings were counted after 45 days of treatment initiation. They were considered alive when there was fresh phloem occurred in both stem and roots, and green tissue on leaf blades. Seedling height was first measured from the new soil surface level to the seedling apex, and then marked immediately. Stem diameter was measured close to the burial surface using a vernier caliper. In the meantime, 15 randomly selected seedlings were dug out in each treatment; roots were picked up as intact as possible from the sandy soil. Taproot lengths were measured, and total leaf area was obtained using a Portable Area Meter (Li-Cor3000A, Lincoln, Nebraska, USA). Finally, plant organs (i.e., leaves, stems and roots) were dried at 80°C and weighed thereafter reaching a constant mass for each seedling in the laboratory.

*Calculations*

The (1) relative height growth rate (RHGR, mm·cm$^{-1}$d$^{-1}$), and (2) relative mass growth rate of seedlings (RMGR, mg·mg$^{-1}$.d$^{-1}$) were calculated according to the following equations (Walck et al., 1999; Zhao et al., 2007):

$$RHGR=\frac{H_2-H_1}{H_1(T_2-T_1)} \quad (1);$$

$$RMGR=\frac{\ln M_2-\ln M_1}{T_2-T_1} \quad (2);$$

where $H_2$ and $H_1$ were seedling heights at the end and beginning (i.e., immediately before sand burial) of the experiment, respectively; $M_2$ or $M_1$ were the total dry biomass of seedlings either after 45 days from study initiation or just before sand burial, respectively; ln was the natural logarithm, and $T_2$-$T_1$ was time from sand burial (i.e., 45 days).

*Statistical analysis*

All data were tested for normality and homogeneity of variance prior to analysis. Data were log-transformed if necessary (Sokal and Rohlf 1995). The effects of experimental sand burial on seedling height, RHGR, plant stem diameter, total leaf area, RMGR, dry biomass and percentage biomass allocation were evaluated by one-way ANOVA. Whenever F tests were significant, Tukey's test was used to compare treatment means at $P<0.05$. All statistical analysis used SPSS 21.0 (SPSS Inc., Chicago, USA), and drawing was made using Origin Pro 9.0 (Origin Lab Corp, USA).

**Results**

*Effects of permanent sand burial on seedling survival*

The effect of sand burial depth on seedling survival was significant ($F_{4,25}$=38.339, $P<0.001$). During the whole study, seedling survival was 100% on the unburied (T0) and partial burial treatments (T33, T67). Simultaneously, seedling survival (84.48±8.8%) was significantly lower in the completely sand burial treatment (T100) than in the control treatment. No seedling survived after burial depth exceeded the overall mean. original height of seedlings (i.e., on T133).

*Changes of morphological seedling traits in response to sand burial*

Seedling height was significantly affected by sand burial depths after 45 days of burial ($F_{3,56}$=139.978, P≤0.001). The highest seedling height was observed in the T33 treatment, which was significantly greater than that in the T67 treatment (Figure 2A;

P<0.05). Height of seedlings in the control treatment was significantly lower than that in the T33 and T67 treatments, but higher than that in the T100 treatment (i.e., 10.66±0.66 cm;Figure 2A. P<0.05).

The relative height growth rate of seedlings (RHGR) was significantly affected ($F_{3,56}$= 286.877; P≤0.001) after 45 days ofsand burial (Figure 2B). Highest (0.057±0.004 mm·cm$^{-1}$d$^{-1}$) and lowest (0.023±0.006 mm·cm$^{-1}$d$^{-1}$) relative growth rates for seedling height were shown in the T33 and T100 treatments, respectively (Figure 2B). The pattern of change with burial depth was similar to that described for seedling height (Figure 2A); values were greater in the control than 100% covered by sand (Figure 2B; P<0.05).

After 45 days from initiation of the study, stem diameter ($F_{3,56}$=26.669, P≤0.001), taproot length ($F_{3,56}$=30.942, P<0.001) and total leaf area ($F_{3,56}$=35.961, P<0.001) of seedlings were also affected by sand burial (Figure 3 A, B, C). Stem diameter was

20% greater (P<0.05) in the T33 than in the T0 treatment (Figure 3A). However, stem diameters were similar in the control and T67 treatments (Figure 3A, P>0.05). Values in the T100 treatment, nevertheless, were 13.4% lower than those in the unburied control (Figure 3A). While taproot length was lowest in the control treatment, it was highest in the T67 treatment (Figure 3B; 35.7% higher than that in the control; P<0.05). The total leaf area of seedlings was significantly greater in the

treatment (Figure 3C).

*Effects of sand burial on biomass growth and relative mass growth rate*

There were significant differences in total seedling biomass ($F_{3,56}$=129.949, P≤0.001) and its component organs [e.g., leaves ($F_{3,56}$=93.965, P<0.001) and roots ($F_{3,56}$=50.474, P=0.002)] after the experiment. The only exception was for seedling stem biomass ($F_{3,56}$=2.017, P=0.122) which was similar in all sand burial treatments (Table 1). Greatest total biomasses were reached in the T33 (369.65±17.27 mg) and T67 treatments (372.50±15.74 mg) (Table 1). Total biomass of seedlings was significantly lower in T100 treatment than in those treatments (Table 1). Patterns shown for the biomasses of leaves and roots among treatments were similar for the total biomass of seedlings (Table 1).

Significant differences were found in allocation of seedling biomass to leaves ($F_{3,56}$=12.841, P<0.001), stems ($F_{3,56}$=27.579,

P≤0.001) and roots ($F_{3,56}$=7.594, P<0.001). On leaves, percentage biomass allocation was greatest in the T33 and T67

treatment, and lowest in the control and T100 treatments (Table 2). Percentage biomass allocation to stems was greatest in the T100 treatment (24.1±1.9%). values on stems was greater in the control than in the T33 and T67 treatments (Table 2).

Finally, percentage biomass allocation to roots showed a slight decreasing trend from the control to T33 and T67 treatments (Table 2); values determined in the T100 treatment for this organ were significantly lower than in the control and T33

treatment (Table 2).

The relative mass growth rate of seedlings was significantly affected by sand burial at the end of experiment ($F_{3,56}$=136.370,

P≤0.001). Greatest relative mass growth rate values of 0.031±0.001 mg・mg$^{-1}$day$^{-1}$ were shown both in the T33 and T67

treatments (Figure 4). These values were significantly greater than those found in the control (Figure 4, P<0.05). Lowest relative mass rates of growth were determined on seedlings grown in the T100 treatment (0.026±0.001 mg·mg$^{-1}$day$^{-1}$; Figure 4).

**Discussion**

*Seedling survivorship in response to permanent sand burial*

In sand land regions, seedlings might be buried at different depths between emergence and the end of the windy season, from late spring to early summer (Chen and Maun, 1999) and this was stimulated in our experiment. After the 45-days, partial burial with sand (to 33% or 67% of their height) did not influence the survival of seedling of sandy elm seedling, as there was no mortality. These results agreed with studies of He et al. (2008), Liu et al. (2008) and Qu et al. (2012), which reported that survivorship of *Artemisia halodendron*, *Corispermum macrocarpum* and *Caragana microphylla* was either maintained or increased by moderate sand burial in Horqin Sandy Land. Survivorship of these shrubs, however, declined among plant species once their seedlings were covered by sand either equal to or more than 100% of their height, and this was also the case for our experiment. Survival decreased sharply by a mean of 15.6%, when seedling were completely buried (to 100% of their height), while deeper sand burial resulted in no survival of sandy elm at all, as seedlings withered and rotted in the soil .This precise threshold offers a clear primary explanation for the absence of sandy elm seedling after relatively deep sand burial. Field survey in recent years has showed that serious land degradation and reduction of vegetation cover has aggravated sand mobility, particularly in the leeward and semi-fixed dunes. Seedlings could successfully complete their periodic recruitments only taking advantage of scarce favorable spatio-temporal chances (Tian et al., 2015; Wu et al., 2013)

Maun (1981, 1997) and Disraeli (1984) also indicated that a certain tolerance sand burial was an effective strategy for survival and subsequent establishment of seedlings in sandy environments. Most seedlings of the grass *Distichlis spicata*

died when completely covered by sand in North America (Brown, 1997; Li et al., 2015), some *Artemisia squarrosum*

seedlings remained alive even though sand burial depths reached 266% of initial seedling height in the Horqin Sandy

Land (Li et al. 2015). Thus, compared with other species, seedling of sandy elm sowed moderate resistance to sand burial.

Harris and Davy (1987) and Perumal and Maun (2006) suggested that plant energy exhaustion and suppression of photosynthesis were implicated in the severely reduced intense radiation and high temperature to some extent. Seedlings of sandy elm have adapted to extreme conditions and previous research has confirmed that sandy elm had a higher transpiration rate and stomatal conductance with lower photosynthesis water-use efficiency and less sensitivity to high temperature and irradiance, compared with other native tree species such as *Malus baccata*, *Prunus padus* and *Pinus sylvestris*, especially in the midday (Park et al., 2012).

*Effects of sand burial on seedling morphological traits*

Sand burial modifies the environment of living plants, forming new microhabitat available for seedling (Disraeli, 1984;

Sun et al., 2014). Plants would be expected to adjust their morphological performances and developments to maximize photosynthetic efficiency and sustain survival (Wang et al., 2014). Our results demonstrated that various seedling morphological traits (i.e., height; stem diameter; taproot length; total leaf area; dry biomass; partitioning of biomass to shoots and roots; RHGR and RMGR ) were increased by partial butial, especially at T33. Thus seedling height of sandy elm was greater in the partially buried than in the unburied and completely buried treatments, which indicated that partial sand burial stimulated stem elongation. This might be explained by the fcat that the processes of growth and elongation, benefiting from improved water maintenance and nutrient uptake in these arid and semi-arid regions (Li et al., 2015). In Horqin Sandy Land, dry sandy layer was reported in the first 5 cm from soil surface level in semi-fixed dunes. This suggests that suitable sand burial depth could be beneficial in reducing soil temperature and keep moisture for root uptake, which are critical to seedlings survival and resource capturing (Niu et al., 2015). Our finding was also consistent with previous research reported from Disraeli (1984), who reported that partial burial stimulated growth of *Ammophila breviligulata* in coastal dunes of northeastern North America. Belcher (1977) also determind that seedling heights of *Rosa rugosa* was higher in the partial than in the unburied and completely continuous sand burial treatments in the desert.

Seedling height growth rate was a critical parameter to determine the speed of growth. The greater RHGR of seedlings in the partial (T33 and T67) than in the unburied and completely buryied treatments (T100) was an indication that partial burial did contribute to a greater seedling height after a 45-day-growth period via accelerating the speed of growth in height. Nevertheless, the greatest seedling growth in height in the T33 treatment came from its greatest RHGR in this than any other treatments. Liu et al. (2008) and Miao et al. (2012) also found that shallow soil burial depths could promote the relative growth rates in height on *Salix gordejevii*, *Artemisia wudanica* and *Artemisia halodendron*. However, there is not universally true as some species(e.g., *Artemisi gmelinii*) have decreased their growth rates as a result of sand burial (Liu et al., 2008). These findings confirmed that the phenotypic response to the degree of sand burial might be species-specific.

Compared with the unburied treatment, partial burial treatments fostered increments in stem diameter and taproot length. Sun et al. (2014a) also showed that seedling diameter and taproot length of *Suaeda salsa* were increased by a partial burial treatment in the coastal marches of the Yellow River estuary, China. Caldwell et al. (1998) found that increases in taproot length were conducive to a greater nutrient and water uptake from deeper soil depths. Total leaf area, however, was either similar or lower in the partial burial than in the unburied treatment. This was in agreement with the results of Liu and Guo (2005), who noted that increasing depth of sand burial decreased the total leaf area of *Caragana intermedia*.

Changes of biomass allocation and relative mass growth rate, which are involved in successful seedling recruitment, have been regarded as the results of variations in plant adaptive strategies in response to environmental changes, (Wu et al., 2013).

Appropriate resource allocation is essential for plant establishment and growth (Bazzaz, 1997). Also, plants may shift resource allocation to minimize the effects of external environmental changes (Maun, 1997; Ni et al., 2015). Numerous findings, especially those on sandy environments, have reported that plants could withstand episodes sand burial by changing biomass allocation. Some species (e.g. *Artemisia ordosia, Elymusfarctus*) may transfer biomass from underground to leaf and stem organs (Brown, 1997; Li et al., 2010), while others (e.g.,*Caragana microphylla, Nitraria sphaerocarpa*) have either maintained or increased biomass allocation to roots (He et al., 2008; Sykes and Wilson, 1990). In our experiment, it was somewhat surprising that no differences were observed in the dry biomass of seedling stems among all sand burial treatments.

However, seedlings showed an increased stem diameter in the T33 sand burial treatment; similar to results of Zhao et al.(2015), this was most likely because of fresh stem had a greater water content in the T33 than in the other treatments.

Additionally, partial burial treatments produced greater dry biomass for leaves, roots and the whole seedlings in comparison with unburied and completely sand burial treatments. Previous studies also determined that 67% burial of seedlings of the shrub *Caragana intermedia* determined a greater biomass allocation to leaves and stems than to roots, compared with values in the unburied control (Xu et al., 2013). Meanwhile, there was a trend for an increasing aboveground (leaf + stem), and a decreasing belowground allocation with increasing burial depth for sandy elm seedlings. Nearly 50% of the total seedling biomass corresponded to leaves. This indicated sufficient leaves are necessary on sandy elm to sustain photosynthesis and maintain evapotranspiration after exposure to high temperature and intense irradiance environments during the growing season (Dulamsuren et al., 2009; Li et al., 2003; Park et al., 2012). The relatively investment in root was slight decreased, which indicated that, on one hand, greater soil moisture availability weakened the dependence on root function, and on the other hand, the plant's need to divert biomass aboveground for light interception and net assimilation rate (Maun, 1994; Sun et al., 2010; Wang et al., 2014).

Relative mass growth rates measure the mean efficiency rate for producing new biomass (Walck et al., 1999). Dalling and Hubbell (2002) showed that seedling growth rate was a better determinant of successful seedling establishment than biomass. In our experiment, relative mass growth rates were higher in the partial than in the other treatments, indicating that moderate sand burials were beneficial for a rapid biomass increase. However, all mass relative growth rates were small compared with those of other plant species (e.g., *Artemisia wudanica, Solidago shortii* and *Solidago. altissima*) in the same area. This suggests that the relative lower biomass accumulation on sandy elm seedlings during the first growing season places these seedlings at a disadvantage when considering soil resource competition and coexistence with other species (Brown, 1997; Liu et al., 2014; Wu et al., 2013). Reduced dry matter accumulation could also have contributed to increased mortality. Although there were striking effects on biomass accumulation and allocation reflecting the plasticity of various morphological traits. Partial sand burial treatments did not change the survivorship, and these treatments facilitate individual seedling growth and population regeneration through phenotylic plasticity or various morphological traits. Our study, however, was conducted in pots. We have to recognize it has limitations in comparison to field studies. Under natural rangeland conditions, some factors (e.g., abrasion of plant tissues by sand grains; grazing by herbivores and granivores)

might reduce or eliminate some of the positive effects of the partial burial treatments (Dulamsuren et al., 2009; Jeffreyt et al.,

2009). Furthermore, we found evidence of allometry of each plant proportion with increasing seedling age in our experiment.

Therefore, more comprehensive studies on physiological and biochemical mechanisms involved on sandy elm seedling survivorship and performance under field, sand burial conditions at different growth stage in future research.

**Conclusion**

Sand burial affected seedling survivorship, growth and biomass allocation of *Ulmus pumila* var. *sabulosa* through phenotypic plasticity of morphological traits. Seedlings of sandy elm showed adaptive responses to moderate sand burial, consistent with its evolution in the sandy environment. Partial sand burial treatment did not influence seedling survivorship, but complete sand burial significantly increased mortality. Compared with the unburied treatment, seedling height, relative height growth rates, taproot length, total biomass and relative mass growth rates were stimulated by partial burial with sand. At the same time, percentage biomass allocation of seedlings was changed. diverting more biomass to aboveground organs (e.g., leaf and stem) to sustain normal photosynthesis and evapo-transpiration. Complete sand burial after seedling emergence, however, inhibited their growth, and even resulted in seedling death. Consequently, burial depths should be controlled by making enclosures and increasing vegetation coverage to facilitate regeneration or re-establishment of sandy elm. The observed variation in all parameters has defined the tolerance of *Ulmus pumila* var. *sabulosa* to sandy environments and its capacity to acclimate them. Hence our research provides a theoretical support for recruitment in sandy sparse elm woodlands.

**Acknowledgements**

This study was supported by National Natural Science Foundation of China (31370706) and the Wulanaodu Desertification Experimental Station of the Institute of Applied Ecology, Chinese Academy of Sciences. We thank Yongming Luo ,Xuehua Li, Quanlai Zhou, Hongmei Wang, Meiyu Jia, Ya Liu and Xu Han for assistance during the experiment. Thanks Authony J. Davy from University of East Anglia for guiding of statistical analysis and language polishing for the original manuscript.

**References**

Bazzaz, F. A.: Allocation of Resources in Plants: State of the Science and Critical Questions. In: Plant Resource Allocation, Academic Press, San Diego, 1997.

Belcher, C. R.: Effect of sand cover on the survival and vigor of *Rosa rugosa* Thunb, Int J Biometeorol, 21, 276-280, 1977.

Bellamy, P. H., Loveland, P. J., Bradley, R. I., Lark, R. M., and Kirk, G. J. D.: Carbon losses from all soils across England and Wales 1978-2003, Nature, 437, 245-248, 2005.

Berendse, F., van Ruijven, J., Jongejans, E., and Keesstra, S.: Loss of Plant Species Diversity Reduces Soil Erosion Resistance, Ecosystems, 18, 881-888, 2015.

Brevik, E. C., Cerdà A., Mataix-Solera, J., Pereg, L., Quinton, J. N., Six, J., and Van Oost, K.: The interdisciplinary nature of SOIL, SOIL, 1, 117-129, 2015.

Brown, J. F.: Effects of Experimental Burial on Survival, Growth, and Resource Allocation of Three Species of Dune Plants, Journal of Ecology, 85, 151-158, 1997.

Cao, C.Y., Jiang, D.M., Teng, X.H., Jiang, Y., Liang, W.J., and Cui, Z.B.: Soil chemical and microbiological properties along a chronosequence of *Caragana microphylla* Lam. plantations in the Horqin sandy land of Northeast China, Applied Soil Ecology, 40, 78-85, 2008.

Cao, C.Y., Jiang, S.Y., Zhang,Y., Zhang, F.X., and Han, X.S.: Spatial variability of soil nutrients and microbiological properties after the establishment of leguminous shrub *Caragana microphylla* Lam. plantation on sand dune in the Horqin Sandy Land of Northeast China, Ecological Engineering, 37, 1467-1475, 2011.

Cerdà A.: The influence of aspect and vegetation on seasonal changes in erosion under rainfall simulation on a clay soil in Spain, Canadian Journal of Soil Science, 78, 321-330, 1998.

Chen, H. and Maun, M. A.: Effects of sand burial depth on seed germination and seedling emergence of *Cirsium pitcheri*, Plant Ecology, 140, 53-60, 1999.

Cheplick, G. P. and Grandstaff, K.: Effects of sand burial on purple sandgrass (*Triplasis purpurea*): the significance of seed
heteromorphism, Plant Ecology, 133, 79-89, 1997.
Disraeli, D. J.: The Effect of Sand Deposits on the Growth and Morphology of Ammophila Breviligulata, Journal of Ecology,
72, 145-154, 1984.
Dulamsuren, C., Hauck, M., Nyambayar, S., Bader, M., Osokhjargal, D., Oyungerel, S., and Leuschner, C.: Performance of
Siberian elm (*Ulmus pumila*) on steppe slopes of the northern Mongolian mountain taiga: Drought stress and herbivory
in mature trees, Environmental and Experimental Botany, 66, 18-24, 2009.
Foley, J. A., DeFries, R., Asner, G. P., Barford, C., Bonan, G., Carpenter, S. R., Chapin, F. S., Coe, M. T., Daily, G. C., Gibbs,
H. K., Helkowski, J. H., Holloway, T., Howard, E. A., Kucharik, C. J., Monfreda, C., Patz, J. A., Prentice, I. C.,
Ramankutty, N., and Snyder, P. K.: Global Consequences of Land Use, Science, 309, 570-574, 2005.
Gabarrón-Galeote, M. A., Martínez-Murillo, J. F., Quesada, M. A., and Ruiz-Sinoga, J. D.: Seasonal changes in the soil
hydrological and erosive response depending on aspect, vegetation type and soil water repellency in different
Mediterranean microenvironments, Solid Earth, 4, 497-509, 2013.
Harris, D. and Davy, A. J.: Seedling Growth in Elymus farctus after Episodes of Burial with Sand, Annals of botany, 60,
587-593, 1987.
He, Y.H., Zhao, H.L., Zhao, X.Y., and Liu, X.P.: Effects of different sand burial depths on growth and biomass allocation in
*Caragana microphylla* seedlings, Arid Land Geography, 31, 701-706, 2008.
Jeffreyt, B., Bobbie, M. M., Johnj, B., Dennisc, G., Robertj, L., and Jhonathane, E.: Sand Abrasion Injury and Biomass
Partitioning in Cotton Seedlings, Agronomy Journal, 101, 1297-1303, 2009.
Jiang, D.M., Liu, Z.M., Cao, C.Y., Kou, Z.W., and Wang, R.Y.: Desertification and Ecological Restoration of Keerqin Sandy
Land, China Environmental Science Press, Beijing, 2003.
Jiang, D.M., Tang, Y., and Busso, C. A.: Effects of vegetation cover on recruitment of *Ulmus pumila* L. in Horqin Sandy
Land, northeastern China, Journal of Arid Land, 6, 343-351, 2013.
Li, J., Qu, H., Zhao, H.L., Zhou, R.L., Yun, J.Y., and Pan, C.C.: Growth and physiological responses of *Agriophyllum*
*squarrosum* to sand burial stress, Journal of Arid Land, 7, 94-100, 2015.
Li, S.L., Werger, M. A., Zuidema, P., Yu, F.-H., and Dong, M.: Seedlings of the semi-shrub *Artemisia ordosica* are resistant
to moderate wind denudation and sand burial in Mu Us sandland, China, Trees, 24, 515-521, 2010.
Li, X.H., Jiang, D.M., Zhou, Q.L., and Oshida, T.: Soil seed bank characteristics beneath an age sequence of *Caragana*
*microphylla* shrubs in the Horqin Sandy Land region of northeastern China, Land Degradation & Development, 25,
236-243, 2014.
Li, Y. G., Jiang, G. M., Niu, S. L., Liu, M. Z., Peng, Y., Yu, S. L., and Gao, L. M.: Gas Exchange and Water Use Efficiency
of Three Native Tree Species in Hunshandak Sandland of China, Photosynthetica, 41, 227-232, 2003.
Liu, B., Liu, Z.M., and Guan, D.X.: Seedling growth variation in response to sand burial in four Artemisia species from
different habitats in the semi-arid dune field, Trees, 22, 41-47, 2008.
Liu, B., Liu, Z.M., Lü, X.T., Maestre, F. T., and Wang, L.X.: Sand burial compensates for the negative effects of erosion on
the dune-building shrub *Artemisia wudanica*, Plant and Soil, 374, 263-273, 2014.
Liu, H. and Guo, K.: The impacts of sand burial on seedling development of Caragana intermedia, Acta Ecologica Sinica, 25,

2550-2555, 2005.

Liu, Z.M., Zhu, J.L., and Deng, X.: Arrival vs. retention of seeds in bare patches in the semi-arid desertified grassland of
Inner Mongolia, northeastern China, Ecological Engineering, 49, 153-159, 2012.

Ma, C. G.: A Provenance Test of White Elm (*Ulmus-Pumila* L) in China, Silvae Genet, 38, 37-44, 1989.

Maun, M. A.: Adaptations enhancing survival and establishment of seedlings on coastal dune systems, Vegetatio, 111, 59-70,
1994.

Maun, M. A.: Adaptations of plants to burial in coastal sand dunes, Canadian Journal of Botany-Revue Canadienne De
Botanique, 76, 713-738, 1997.

Maun, M. A. and Lapierre, J.: Effects of Burial by Sand on Seed Germination and Seedling Emergence of Four Dune Species,
American Journal of Botany, 73, 450-455, 1986.

Miao, R.H., Jiang, D.M., Musa, A., Zhou, Q.L., Guo, M.X., and Wang, Y.C.: Effectiveness of shrub planting and grazing
exclusion on degraded sandy grassland restoration in Horqin sandy land in Inner Mongolia, Ecological Engineering, 74,
164-173, 2014.

Miner, B. G., Sultan, S. E., Morgan, S. G., Padilla, D. K., and Relyea, R. A.: Ecological consequences of phenotypic
plasticity, Trends in Ecology & Evolution, 20, 685-692, 2005.

Ni, J., Luo, D. H., Xia, J., Zhang, Z. H., and Hu, G.: Vegetation in karst terrain of southwestern China allocates more
biomass to roots, Solid Earth, 6, 799-810, 2015.

Niu, C. Y., Musa, A., and Liu, Y.: Analysis of soil moisture condition under different land uses in the arid region of Horqin
sandy land, northern China, Solid Earth, 6, 1157-1167, 2015.

Park, Y. D., Lee, D. K., Batkhuu, N. O., Tsogtbaatar, J., Combalicer, M. S., Go, E., Park, and Su, Y. W.: Woody Species
Variations in Biomass Allocation, Photosynthetic WUE and Carbon Isotope Composition under Natural Drought
Condition in Mongolia, Journal of Environmental Science & Management, 2012. 29-37, 2012.

Perumal, V. J. and Maun, M. A.: Ecophysiological response of dune species to experimental burial under field and controlled
conditions, Plant Ecology, 184, 89-104, 2006.

Qian, J.Q., Liu, Z.M., Hatier, J. H. B., and Liu, B.: The Vertical Distribution of Soil Seed Bank and Its Restoration
Implication in an Active Sand Dune of Northeastern Inner Mongolia, China, Land Degradation & Development, 2015.
2015.

Qiu, S.W.: Study on the formation and evolution of Horqin Sandy Land, Scientia Geographica Sinica, 9, 317-328, 1989.

Qu, H., Zhao, H.L., Zhou, R.L., Zuo, X.A., Luo, Y., Wang, J., and Barron, J. O.: Effects of sand burial on the survival and
physiology of three psammophytes of Northern China, African Journal of Biotechnology, 11, 4518-4529, 2012.

Schlütz, F., Dulamsuren, C., Wieckowska, M., Mühlenberg, M., and Hauck, M.: Late Holocene vegetation history suggests
natural origin of steppes in the northern Mongolian mountain taiga, Palaeogeography, Palaeoclimatology, Palaeoecology,
261, 203-217, 2008.

Shi, L., Zhang, Z. J., Zhang, C. Y., and Zhang, J. Z.: Effects of sand burial on survival, growth, gas exchange and biomass
allocation of *Ulmus pumila* seedlings in the Hunshandak Sandland, China, Annals of botany, 94, 553-560, 2004.

Smith, P., Cotrufo, M. F., Rumpel, C., Paustian, K., Kuikman, P. J., Elliott, J. A., McDowell, R., Griffiths, R. I., Asakawa, S.,
Bustamante, M., House, J. I., Sobocká J., Harper, R., Pan, G., West, P. C., Gerber, J. S., Clark, J. M., Adhya, T., Scholes,

R. J., and Scholes, M. C.: Biogeochemical cycles and biodiversity as key drivers of ecosystem services provided by soils,
SOIL, 1, 665-685, 2015.

Sun, Z.G., Mou, XJ.., Lin, G.H., Wang, L.L., Song, H.L., and Jiang, H.: Effects of sediment burial disturbance on seedling
survival and growth of *Suaeda salsa* in the tidal wetland of the Yellow River estuary, Plant and Soil, 337, 457-468, 2010.

Sun, Z.G., Song, H.L., Sun, W.G., and Sun, J.K.: Effects of continual burial by sediment on morphological traits and dry
mass allocation of *Suaeda salsa* seedlings in the Yellow River estuary: An experimental study, Ecological Engineering,
68, 176-183, 2014.

Sykes, M. T. and Wilson, J. B.: An experimental investigation into the response of New Zealand sand dune species to
different depths of burial by sand, Acta Botanica Neerlandica, 39, 171-181, 1990.

Tang, J., Busso, C., Jiang, D.M., Wang, Y.C., Wu, D.F., Musa, A., Miao, R.H., and Miao, C.P.: Seed Burial Depth and Soil
Water Content Affect Seedling Emergence and Growth of *Ulmus pumila* var. sabulosa in the Horqin Sandy Land,
Sustainability, 8, 68, 2016.

Tang, J., Jiang, D.M., and Wang, Y.C.: A review on the process of seed-seedling regeneration of *Ulmus pumila* in sparse
forest grassland, Chinese Journal of Ecology, 33, 1114-1120, 2014.

Tang, Y., Jiang, D.M., and Lü, X.T.: Effects of Exclosure Management on Elm (*Ulmus Pumila*) Recruitment in Horqin Sandy
Land, Northeastern China, Arid Land Research and Management, 28, 109-117, 2013.

Tian, F.P., Liu, Y., Wu, G.L., and Shi, S.L.: Seedling recruitment of forb species under experimental microhabitats in alpine
grassland, Pakistan Journal of Botany, 47, 2127-2134, 2015.

Verheijen, F. G. A., Jones, R. J. A., Rickson, R. J., and Smith, C. J.: Tolerable versus actual soil erosion rates in Europe,
Earth-Science Reviews, 94, 23-38, 2009.

Walck, J. L., Baskin, J. M., and Baskin, C. C.: Relative competitive abilities and growth characteristics of a narrowly
endemic and a geographically widespread Solidago species (Asteraceae), American Journal of Botany, 86, 820-828,
1999.

Wang, D., Zhu, Y. J., Wu, G. L., and Feng, J.: Seedling performance within eight different seed-size alpine forbs under
experimentation with irradiance and nutrient gradients, Pakistan Journal of Botany, 46, 1261-1268, 2014.

Wang, J., Zhou, R.L., Zhao, H.L., Zhao, Y., and Hou, Y.: Growth and physiological adaptation of *Messerschmidia sibirica* to
sand burial on coastal sandy, Acta Ecologica Sinica, 32, 4291-4299, 2012.

Wang, K.B., Deng, L., Ren, Z.P., Li, J.P., and Shangguan, Z.: Grazing exclusion significantly improves grassland ecosystem
C and N pools in a desert steppe of Northwest China, Catena, 137, 441-448, 2016.

Wu, G. L., Feng, J., Shi, Z. H., and Du, G. Z.: Plant seedling performance traits impact on successful recruitment in various
microhabitats for five Alpine Saussurea species, Pakistan Journal of Botany, 45, 61-71, 2013.

Xu, L., Huber, H., During, H. J., Dong, M., and Anten, N. P. R.: Intraspecific variation of a desert shrub species in
phenotypic plasticity in response to sand burial, New Phytologist, 199, 991-1000, 2013.

Yan, Q.L., Liu, Z.M., Zhu, J.J., Luo, Y.M., Wang, H.M., and Jiang, D.: Structure, Pattern and Mechanisms of Formation of
Seed Banks in Sand Dune Systems in Northeastern Inner Mongolia, China, Plant & Soil, 277, 175-184, 2005.

Yang, H.L., Cao, Z.P., Dong, M., Ye, Y.Z.., and Huang, z.: Effects of sand burying on caryopsis germination and seedling
growth of *Bromus inermis* Leyss, Chinese Journal of Applied Ecology, 18, 2438-2443, 2007.

Yang, L.M., Zhou, G.S., Wang, G.H., and Wang, Y.H.: Effect of human activities on soil environment and plant species diversity of elm sparse woods, The Journal of Applied Ecology, 14, 321-325, 2003.

Zhang, T.H., Zhao, H.L., Li, S.G., Li, F.R., Shirato, Y., Ohkuro, T., and Taniyama, I.: A comparison of different measures for stabilizing moving sand dunes in the Horqin Sandy Land of Inner Mongolia, China, Journal of Arid Environments, 58, 203-214, 2004.

Zhao, H.L., Li, J., Zhou, R.L., Qu, H., Yun, J.Y., and Pan, C.C.: Effects of Sand Burial on Growth Properties of *Pinus sylvestnis* var.mongolica, Journal of Desert Research, 35, 60-65, 2015.

Zhao, W. Z., Li, Q. Y., and Fang, H. Y.: Effects of sand burial disturbance on seedling growth of *Nitraria sphaerocarpa*, Plant and Soil, 295, 95-102, 2007.

**Table legends**

Table. 1    Dry biomass of leaves, stems and roots for seedlings of Ulmus pumila var.sabolusa exposed to various sand burial treatments during a 45-day-growth period. These treatments included sand burial of seedlings to a depth equivalent to 33 (T33), 67 (T67) or 100% (T100) of the mean seedling height at the initiation of the study. Each histogram is the mean ±1 S.E. of n=15. Different letters in a row among seed burial depths are significantly different at P<0.05.

Table. 2    Percentage biomass allocation to leaves, stems and roots on seedlings of *Ulmus pumila* var.*sabolusa* expose to various sand burial treatments during a 45-day-growth period. These treatments included sand burial of seedlings to a depth equivalent to 33 (T33), 67 (T67) or 100% (T100) of the mean seedling height at the initiation of the study. Each histogram is the mean ±1 S.E. of n=15. Different letters above histograms among seed burial depths are significantly different at P<0.05.

**Figure legends**

Fig. 1    The geographic location of study area in Horqin Sandy Land, China.

Fig.2    Seedling height and relative growth rate for height (RHGR) of *Ulmus pumila* var.*sabolusa* exposed to various sand burial treatments during a 45-day- growth period. These treatments included sand burial of seedlings to a depth equivalent to 33 (T33), 67 (T67) or 100% (T100) of the mean seedling height at the initiation of the study (see the Material and Methods Section for further details). Each histogram is the mean ±1 S.E. of n=15. Different letters above histograms among seed burial depths are significantly different at P<0.05.

Fig.3.    Stem diameter, taproot length and total leaf area on seedlings of *Ulmus pumila* var. *sabolusa* exposed to various sand burial treatments during a 45-day-growth period. These treatments included sand burial of seedlings to a depth equivalent to 33 (T33), 67 (T67)

or 100% (T100) of the mean seedling height at the initiation of the study. Each histogram is the mean ±1 S.E. of n=15. Different letters above histograms among seed burial depths are significantly different at $P<0.05$.

**Fig.4**    Relative mass growth rates (RMGR) on seedlings of *Ulmus pumila* var. *sabulosa* exposed to various sand burial treatments during a 45-day-growth period. These treatments included sand burial of seedlings to a depth equivalent to 33 (T33), 67 (T67) or 100% (T100) of the mean seedling height at the initiation of the study. Each histogram is the mean ± 1 S.E. of n=15. Different letters above histograms among seed burial depths are significantly different at $P<0.05$.

Table 1

| Treatment | Stem mg | Root mg | Leaf mg | Total mg |
|---|---|---|---|---|
| T0 | 66.62±3.49a | 91.7±7.51b | 160.63±9.15b | 318.95±14.85b |
| T33 | 69.52±7.76a | 104.59±9.89c | 195.55±11.54c | 369.65±17.27c |
| T67 | 68.69±4.05a | 101.64±6.87c | 200.17±14.73c | 372.5±15.74c |
| T100 | 65.51±3.52a | 69.61±9.84a | 137.55±12.43a | 272.67±16.42a |

| Treatment | Stem % | Root % | Leaf % |
|---|---|---|---|
| T0 | 20.9±1.4b | 28.8±1.6b | 50.3±1.4b |
| T33 | 18.8±1.4b. | 28.3±1.4b | 52.9±1.6c |
| T67 | 18.5±1.3a | 27.3±1.7ab | 54.2±2.4c |
| T100 | 24.1±2.4c | 25.5±2.4c | 50.4±2.4c |

[Figure]


[Figure]

Figure 1

[Figure]

Figure 2

[Figure]

Figure 3

[Figure]

[Figure]

Figure 4

---

## Author Comment (AC3) · 26 May 2016

RC1: L31 "transferring more biomass to aboveground rather than belowground parts." It is interesting, but not be well supported. I suggested to compare the above ground biomass (i.e. stem+leaf) and below ground biomass (i.e. root) respectively. The root biomass was reported in the previous version, but the mean and s.d. values were not shown in the results part. AC1: We have changed the way of expression in the abstract to make it clear in the revised manuscript.

RC2: L109 Before the start of burial treatment experiment, Is the number of seedlings

in each pot same with each other? What is the number of seedlings in each pot? AC2: Yes, twenty days after sowing, 8 to 12 seedlings emerged; eight similar seedlings were retained in each pot, and the rest were removed.

RC3: The second degree of freedom in F test could not be 235 in the experiment described in current status. Please verify it! AC3: We have corrected it in the revised manuscript.

RC4: L147 "Height of seedlings in the control treatment of 16.28cm was significantly lower than that in the T33". The mean value is not enough. The s.d value should be reported. L149 The mean and s.d. values of height growth rate should be reported. L155 The mean and s.d. values of stem diameter should be reported. If it is nessesary, the mean and s.d. values of other indices should be reported. It is helpful for evaluating the effects of sand burial on the growth of elm seedlings. AC4: We have added the mean and s.d. values in the part of results.

For the technical corrections, we have checked according to the reviewer's requests. In addition, three skilled professors helped to revise and polish this revised manuscript. Thank for your valuable suggestions for improving our manuscript.

Please also note the supplement to this comment:
http://www.solid-earth-discuss.net/se-2016-55/se-2016-55-AC3-supplement.pdf

**Supplement:**

|----------------|---------------------------------------------------------------------------------------------------------------------------------------------------------------------------------------------------------------------------------------------------------------------------------------------------------------------|-------------------------------------------------------------------|
| 2              |                                                                                                                                                                                                                                                                                                                     |                                                                   |
| 5              | Experimental sand burial affects seedling survivorship, morphological traits and                                                                                                                                                                                                                                    | 删除的内容: r                                                   |
| 6              | biomass allocation of Ulmus pumila var. sabo/usa in Horqin Sandy Land, China                                                                                                                                                                                                                                 | 删除的内容: e                                                   |
| 8              | Jiao Tang 1,2, , Carlos Alberto Busso 3 , Deming Jiang 1* , Ala Musa 1 , Dafu Wu 4 , Yongcui Wang 1 , Chunping Miao 1,2                                                                                                                | 删除的内容: t                                                   |
| 11
12
13 | 1 Institute of Applied Ecology, Chinese Academy of Sciences, Shenyang, 110016, China;
2 University of Chinese Academy of Sciences, Beijing, 100048, China;
3 Departmente de Agronom & CERZOS (CONICET: Conseio Nacional de Investigaciones Cient ficas y Tecnol égicas de la | 带格式的: 字体: (中文) 宋体, 非
加粗, 不对齐到网格
| 14             | República Argentina). Universidad Nacional del Sur. San Andr és 800. 8000 Bah á Blanca. Argentina:                                                                                                                                                                                                                  | 删除的内容: the                                                        |
| 16             | 4 Department of Resource and Environment, Henan Institute of Science and Technology, Xinxiang,453003, China                                                                                                                                                                                              | 删除的内容: with                                                |
| 19
20       | Correspondence to: Deming Jiang ( jiangdeming2016@163 com)                                                                                                                                                                                                                                                   | 删除的内容:)                                                           |
| 22             | Abstract, As a native tree species, Ulmus pumila var. sabolusa (sandy elm) is widely distributed in Horqin Sandy Land,                                                                                                                                                                                              | 删除的内容:                                                            |
| 23             | China . However, seedlings of this species have to withstand various depths of sand burial after emergence because of                                                                                                                                                                                        | 一
| 24             | increasing soil degradation, which is mainly caused by overgrazing and climate change. An experiment was conducted to                                                                                                                                                                                               |                                                                   |
| 25
26       | evaluate the changes in its survivorship , morphological traits and biomass allocation when seedlings buried at different                                                                                                                                                                      | 则除始由效。                                                            |
| 20             | build repuis the results showed that partial sand burial treatments (i.e. less than 67% burial) did not reduce seedling survivorship                                                                                                                                                                                |                                                                   |
| 27             | which still reached 100%. However, seedling mortality increased when sand burial was equal to or greater than 100% Jn                                                                                                                                                                                               | 删除的内容: in comparison with control treatment                       |
| 29             | comparison with the control treatment, seedling height and stem diameter increased at least by 6 and 14% with partial burial,                                                                                                                                                                                       | 副险的内容, Whilet a                                                   |
| 30             | respectively. In the meantime, seeding taproot length, total biomass, and relative mass growth rates at least enhanced by 10%,                                                                                                                                                                                      |                                                                   |
| 31             | 15.6%, and 27.6%, respectively, with the partial sand burial treatment. Furthermore, sand burial decreased total leaf area and                                                                                                                                                                                      | 删陈的内容: on                                                         |
| 32             | changed biomass allocation in seedlings, partitioning more biomass to aboveground organs (eg., leaves and stems) and less                                                                                                                                                                                           | 删除的内容: transferring                                               |
| 33             | to belowground parts (roots). Complete sand burial after seedling emergence inhibited its re-emergence and growth, even                                                                                                                                                                                             | 删除的内容: rather                                                     |
| 34             | leading to its death. Our findings indicated that seedlings of sandy elm had some resistance to partial sand burial and were                                                                                                                                                                                        | 删除的内容: a certain                                                  |

| 60       | adapted to sandy environments from an evolutionary perspective. The negative effects of excessive sand burial after seedling           | _                 | 删除的内容: acclimated                                     |
|----------|----------------------------------------------------------------------------------------------------------------------------------------|-------------------|-------------------------------------------------------|
| 61       | emergence might help to understand failures in recruitment of sparse elm in the study region.                                          |                   | 删除的内容: common                                         |
| 63       | Keywords: sand burial, seedling, sandy elm, morphological, biomass allocation, Horqin Sandy Land                                       |                   | 删除的内容: is                                             |
| 64
CF | In the Justice                                                                                                                         | $\langle \rangle$ | 删除的内容: accretion                                      |
| 05       | Introduction                                                                                                                           |                   | 删除的内容:s                                               |
| 66       | Horqin Sandy Land, shaped in the middle the Pleistocene period, is located in the southeast of the Mongolia Plateau, China             |                   | 删除的内容:1                                               |
| 67       | (Qiu, 1989). Because of climatic changes (rainfall distribution and global warming) and excessive human disturbances (i.e.,            |                   |                                                       |
| 68       | over-utilization of renewable natural resources), vegetation degradation and land desertification have become more obvious             |                   |                                                       |
| 69       | in the past 50 years (Cao et al., 2008; Jiang et al., 2003; Zhang et al., 2004). Sand moves fast in the horizontal or vertical         |                   |                                                       |
| 70       | space because of the effects of strong winds during spring and summer, leading to different burial depths, which might range           |                   |                                                       |
| 71       | from 0.5cm to 56.0cm (Liu et al., 2014).                                                                                               | /                 | 删除的内容: .                                              |
| 72       | Soil genesis is the pivotal process that determines the evolution of the soil system, and offers services and resources to             | /                 | 删除的内容: s                                       |
|          | For genesis is the proved process that determines the production of the son system, and one s ervices and resources to y |                   | 删除的内容:, especially                                    |
| 73       | mankind (Berendse et al., 2015; Niu et al., 2015). Simultaneously, disturbances (such as land-use intensification and                  |                   | 删除的内容: The e                                          |
| 74       | overgrazing) have a profound impact on the soil genesis process because of the increasing population and consumption                   | /                 | 删除的内容: s                                       |
| 75       | (Brevik et al., 2015; Verheijen et al., 2009; Wang et al., 2016). Excessive human interferences change soil hydrological,              |                   | 删除的内容: primary pre                                    |
| 76       | geochemical and biological cycles, inducing serious land degradation such as acidification, salinization and desertification           |                   | 删除的内容: land                                           |
| 77       | (Bellamy et al., 2005; Foley et al., 2005; Gabarr ón-Galeote et al., 2013; Smith et al., 2015). It is well known that vegetation       |                   | 删除的内容: the                                            |
| 78       | plays an important role in controlling soil genesis and degradation in fragile ecosystems such as estuarine, desert and sandy          |                   | 删除的内容: (eg.                                    |
| 79       | lands (Berendse et al., 2015; Cerd à 1998; Miao et al., 2014). Moreover, vegetation functional traits (e.g., morphology and            |                   | 删除的内容: And their s                                    |
| 80       | establishment) in response to environmental stress (e.g., nutrient deficiency, water deficit, high irradiance, extreme                 |                   | growth is always restricte
and harsh environments. |
|          | .,                                                                                                                                     |                   |                                                       |

| 削除的内容:                               |
|--------------------------------------|
| 削除的内容: s                      |
| 削除的内容: , especially           |
| 削除的内容: The e                  |
| 削除的内容: s                      |
| 削除的内容: to                     |
| 削除的内容: primary presented      |
| 削除的内容: land                   |
| 削除的内容: the                           |
| 削除的内容: the                           |
| 削除的内容: (eg.                   |
|                                      |
| 削除的内容: And their survival and |
| rowth is always restricted by barren |

ıу bу d harsh environments.

| 103 | temperatures) might be important adaptive life history adaptive strategy (Miner et al., 2005; Wang et al., 2014). Plants           |
|-----|------------------------------------------------------------------------------------------------------------------------------------|
| 104 | usually have to face a trade-off between survival and growth in response to environmental changes by regulating their              |
| 105 | phenotypic plasticity (e.g., biomass allocation, relative growth rate) and/or physiological traits (e.g., antioxidant enzyme       |
| 106 | activities, membrane permeability, contents of osmotic substance) (Li et al., 2015; Qu et al., 2012; Tian et al., 2015; Wu et al., |
| 107 | 2013)                                                                                                                              |
| 108 | In sandy ecosystems, instability of the soil surface is one of the most damaging factors to biological activity. Furthermore, a    |
| 109 | sparse vegetation cover, and a loose soil texture are, highly susceptible to sand movements (Liu and Guo, 2005; Yan et al.,        |
| 110 | 2005). Sand movement, the most direct evidence caused by land degradation, is regarded as a selective force determining.           |
| 111 | colonization, establishment and establishment of vegetation (Maun, 1994; Maun and Lapierre, 1986). Plants might             |
| 112 | respond differently to various degrees of sand burial, and evolve different regenerative adaptations during the periods of soil    |
| 113 | seed bank formation, seed germination, and seedling emergence and development (Li et al., 2014; Qian et al., 2015; Tang et         |
| 114 | al., 2016).                                                                                                                        |
| 115 | As an indigenous tree species, Ulmus pumila var. sabolusa (sandy elm) has been widely distributed in the leeward slope of          |
| 116 | fixed and semi-fixed sand dunes and became the main component of sparse woodlands of the Horqin Sandy Land (Jiang et               |
| 117 | al., 2013; Tang et al., 2014; Tang et al., 2013). Since prehistoric times, it has been closely relate to human life, providing     |
| 118 | hardwood for farming tools and furniture, fuel for nomad, and fodder from its tender leaves, young fruits and edible               |
| 119 | bark (Ma, 1989; Schlütz et al., 2008). In addition, the sparse-elm woodlands, not only offered shelter for wildlife and            |
| 120 | domestic animals and a suitable environment for psammophytes; but also protected soil from wind erosion and burial,                |

| 144 | providing a very important ecological and social function in these arid and semi-arid lands (Yang et al., 2003). Despite             |
|-----|--------------------------------------------------------------------------------------------------------------------------------------|
| 145 | we realize of the effects of sand burial on establishment of sandy elm, much of our comprehension and recognition to date            |
| 146 | come from ocular observations at the field rather than from controllable experiments (Maun, 1997). For example, we                   |
| 147 | observed that even though many plant of non-dormant, dispersed seeds germinated and seedling emergence occurs in the late            |
| 148 | spring, few surviving seedlings were detected in the following field surveys in the degraded sparse woodlands. This                  |
| 149 | phenomenon is hampering its recruitment and will have negative effects on future community structure of these woodland               |
| 150 | ecosystems                                                                                                                           |
| 151 | Studies on the effects of sand burial have been widely reported in the field of seedlings survival Belcher, 1977; Cheplick           |
| 152 | and Grandstaff, 1997; Harris and Davy, 1987; Li et al., 2015; Liu et al., 2008; Perumal and Maun, 2006), physiological               |
| 153 | characteristics (Shi et al., 2004; Wang et al., 2012; Zhao et al., 2015), and reproductive strategies (Liu et al., 2014; Sun et al., |
| 154 | 2014; Zhao et al., 2007) of coastal marshes and wetlands plants. In general, it appears to be a threshold sand burial depth for      |
| 155 | each plant species to matain vigor and the subsequent sustain growth (Maun, 1997). Below that burial level, plant emergence          |
| 156 | and development have been promoted by increasing sand burial depth (Qu et al., 2012; Yang et al., 2007), Once the threshold          |
| 157 | have been exceeded, a deterioration of seedling vigor and reduced growth has occur, and even leaded to seedling death                |
| 158 | Maun, 1997; Maun and Lapierre, 1986). However, no research has been conducted on the effects of continual sand                       |
| 159 | accumulation on sandy elm seedlings after emergence to date, because of the limited area of sparse elm woodland. So we               |
| 160 | investigated the effects of experimental sand burial on seedling survivorship and growth of sandy elm. The main objectives           |
| 161 | of this study were: (1) to evaluate the effect of sand burial on seedling survivorship: (2) to access the changes on seedling        |

| 世校式的・之休颜色・文字                          | 1        |
|---------------------------------------|----------|
|                                       |          |
| 市相关的                                  |   |
| 删除的内容: n increasing numb              | er of    |
| 一 一 一 一 一 一 一 一 一 一 一 一 一 一 一 一 一 一 一 |          |
|                                       |   |
|                                       | rs (A    |
| (现代吗已史以                               |          |
| 删除的内容:rea ofpars                      | e elm    |
|                                       | l |

| 220 | morphological traits and biomass allocation in response to sand burial, and (3) to explain the failure of sandy elm             |                           |                              |
|-----|---------------------------------------------------------------------------------------------------------------------------------|---------------------------|------------------------------|
| 221 | regeneration, and provide a theoretical basis for achieving a successful recruitment and vegetation establishment on sandy      |                           | 删除的内容: reconstruction        |
| 222 | elm woodlands.                                                                                                                  |                           |                              |
| 223 |                                                                                                                                 |                           |                              |
| 224 | Materials and methods                                                                                                           |                           |                              |
| 225 | Study site                                                                                                                      |                           |                              |
| 226 | The experiment was conducted at the Wulanaodu Desertification Experimental Station of the Institute of Applied Ecology,         |                           | 删除的内容: U'             |
| 227 | Chinese Academy of Sciences (43_02'_N,119_39'_E, 480_m a.s.l) Jocated in western Horqin Sandy Land, China (Figure 1).           |                           | 删除的内容: , where        |
| 228 | This site experiences, temperate continental climate. Mean annual temperature and precipitation are 7.3°C and 315_mm,           |                           | 删除的内容: belongs to            |
| 229 | respectively. Almost 75% of precipitation occurs from June to September during the growing season. Annual average wind          |                           | 删除的内容: during                |
| 230 | speed is 4.4 m s -1 ; the windy season is from March to June (Liu et al., 2012; Miao et al., 2014). The landscape is | $\bigwedge$               | 删除的内容: in                    |
| 221 | abaracterized by sparse woodlands and dunes and lowland areas. The dominant soils are corplian soils, and major plant           | $\langle \rangle$         | 删除的内容: (Figure 1)
| 251 | characterized by sparse woodrands, sand duries and rowrand areas. The uprimiant sons are actorian sons, and major prant         | $\swarrow$                | 删除的内容: /s             |
| 232 | species include some shrubs (e.g., Salix gordejevii and Caragana microphylla,), and perennial and annual herbs (e.g.,           | $\langle \rangle \rangle$ | 删除的内容: D                     |
| 233 | Bassiadasyphylla, Agriophyllum squarrosum, (Cao et al., 2011).                                                                  | $\left  \right\rangle$    | 删除的内容: sandy                 |
| 224 | Experimental methods                                                                                                            | N,                        | 删除的内容: L                     |
| 234 | Experimental methods                                                                                                            | $\sim$                    | 删除的内容: L                     |
| 235 | In mid-May 2015, sandy elm seeds were first collected from multiple, mature individuals and then mixed altogether. After        |                           | 删除的内容 : L             |
| 236 | careful selection, uniform and intact seeds_were chosen and sowed in plastic pots (45_cm diameter, 30_cm height). Sandy soil    |                           |                              |
| 237 | was taken from nearby woodlands, and it was sieved to remove debris and branches. All seeds were covered by sand to a           |                           |                              |

| 252 | depth of 0.5-1.0_cm. In a parallel study, we found that that depth was the most suitable promoting the greatest percentage and       |           | 删除的内容: for having                         |
|-----|--------------------------------------------------------------------------------------------------------------------------------------|-----------|-------------------------------------------|
| 253 | speed of seedling emergence for sandy elm_(Tang et al., 2016). Holes in, the bottom of the pots were covered with nylon              |           | 刪除的内容: at                                 |
| 254 | mesh to prevent soil loss, while allowing drainage of water, All pots were watered every three days to keep the soil moist.          |           | 删除的内容: at the same time            |
| 255 | Twenty days after sowing, 8 to 12 seedlings emerged; eight similar seedlings were retained in each pot, and the rest were            | _         | 删除的内容 : left                       |
| 256 | removed. Mean seedling height (5.4±0.5_cm) was obtained after measuring the height of each seedling in every pot.                    |           | 删除的内容: was                         |
| 257 | Afterwards, seedlings were experimentally buried to either 0_(T0, no burial, control treatment) or 33% (T33; 1.8_cm), 67%            |           |                                           |
| 258 | (T67; 3.6_cm), 100% (T100; 5.4_cm) or 133% (T133; 7.2_cm) soil depth of the original. overall mean seedling height. For,             |           | 删除的内容: With                        |
| 259 | this purpose, sandy soil was added to the pots according to the different burial depths. Each seedling was kept vertical while       |           |                                           |
| 260 | buried. Six replicates were used per treatment, so there was, a total of 30 pots in this experiment. Meanwhile, 15 randomly          |           | 删除的内容: ere                                |
| 261 | selected seedlings were harvested to determine the original measurements for growth analysis before sand burial.                     |           |                                           |
| 262 | Surviving seedlings were counted after 45 days of treatment initiation. They were considered alive when there was fresh              | $\langle$ | 删除的内容: treatment started of               |
| 263 | phloem occurred in both stem and roots, and green tissue on leaf blades. Seedling height was first measured from the new             |           | 删除的内容: in case that
| 264 | soil surface level to the seedling apex, and then marked immediately. Stem diameter was measured close to the burial surface         | $\sum$    | 删除的内容: in                                 |
| 265 | using a vernier caliper. In the meantime, 15 randomly selected seedlings were dug out in each treatment; roots were picked           |           | 删除的内容: Immediately after burial, s |
| 266 | up as intact as possible from the sandy soil. Taproot lengths were measured, and total leaf area was obtained using a Portable       |           | 删除的内容 : M                          |
| 267 | Area Meter (Li-Cor3000A, Lincoln, Nebraska, USA). Finally, plant organs ( i.e. , leaves, stems and roots) were dried at 80°C. |           |                                           |
| 268 | and weighed thereafter reaching a constant mass for each seedling in the laboratory.                                                 |           |                                           |
| 269 | Calculations                                                                                                                         |           |                                           |

| 284 | The (1) relative height growth rate (RHGR, mm_ $cm^{-1}d^{-1}$ ), and (2) relative mass growth rate of seedlings (RMGR, mg_                             |                 | 删除的内容:                                                                            |
|-----|---------------------------------------------------------------------------------------------------------------------------------------------------------|-----------------|-----------------------------------------------------------------------------------|
| 285 | $mg^{-1}$ .d -1 ) were calculated according to the following equations (Walck et al., 1999; Zhao et al., 2007):                              |                 | 刪除的内容:.                                                                           |
| 286 | $RHGR = \frac{H_2 - H_1}{H_1(T_2 - T_1)} $ (1);                                                                                                         |                 | 带格式的:字体:(默认) Times         New Roman         带格式的:字体:(默认) Times         New Roman |
| 287 | $R\underline{M}GR = \frac{\ln M_2 - \ln M_1}{T_2 - T_1} $ (2);                                                                                          |                 | with Nonalin
| 288 | where H 2 and H 1 were seedling heights at the end and beginning (i.e., immediately before sand burial) of the experiment, |                 | 删除的内容: initiation                                                                 |
| 289 | respectively; $M_2$ or $M_1$ were the total dry biomass of seedlings either after 45 days from study initiation or just before sand                     |                 |                                                                                   |
| 290 | burial, respectively; In was the natural logarithm, and T 2 -T 1 was time from sand burial (i.e., 45 days).                       |                 |                                                                                   |
| 291 | Statistical analysis                                                                                                                                    |                 |                                                                                   |
| 292 | All data were tested for normality and homogeneity of variance prior to analysis. Data were log-transformed if necessary                                |                 |                                                                                   |
| 293 | (Sokal and Rohlf 1995). The effects of experimental sand burial on seedling height, RHGR, plant stem diameter, total leaf                               |                 |                                                                                   |
| 294 | area, RMGR, dry biomass and percentage biomass allocation were evaluated by one-way ANOVA. Whenever F tests were                                        |                 |                                                                                   |
| 295 | significant, Tukey's test was used to compare treatment means at P<0.05. All statistical analysis used SPSS 21.0 (SPSS Inc.,                            |                 |                                                                                   |
| 296 | Chicago, USA), and drawing was made using Origin Pro 9.0 (Origin Lab Corp, USA).                                                                        |                 |                                                                                   |
| 297 |                                                                                                                                                         |                 |                                                                                   |
| 298 | Results                                                                                                                                                 |                 | 带格式的: 字体:(默认)Times
New Roman                                            |
| 299 | Effects of permanent sand burial on seedling survival                                                                                            |                 | 删除的内容: continual                                                                  |
| 300 | The effect of sand burial depth on seedling survival was significant (F4,25,=38.339, P≤0.001). During the whole study,                                  | /               | 删除的内容: 235                                                                        |
| 301 | seedling survival was 100% on the unburied (T0) and partial burial treatments (T33, T67). Simultaneously, seedling survival                             |                 | 删除的内容:0                                                                           |

| 309 | (84.48±8.8%) was significantly lower in the completely sand burial treatment (T100) than in the control treatment. No                                                                                                   |
|-----|-------------------------------------------------------------------------------------------------------------------------------------------------------------------------------------------------------------------------|
| 310 | seedling survived after burial depth exceeded the overall mean, original height of seedlings (i.e., on T133).                                                                                                    |
| 311 | Changes of morphological seedling traits in response to sand burial                                                                                                                                                     |
| 312 | Seedling height was significantly affected by sand burial depths after 45 days of burial ( $F_{3,56}$ =139.978, P $\leq$ 0.001). The highest                                                                            |
| 313 | seedling height was observed in the T33 treatment, which was significantly greater than that in the T67 treatment (Figure 2A;                                                                                           |
| 314 | P<0.05). Height of seedlings in the control treatment was significantly lower than that in the T33 and T67 treatments, but                                                                                              |
| 315 | higher than that in the T100 treatment (i.e., $10.66 \pm 0.66$ cm; Figure 2A, P<0.05).                                                                                                                                  |
| 316 | The relative height growth rate of seedlings (RHGR) was significantly affected ( $F_{3,56}$ = 286.877; P $\leq$ 0.00 1 ) after 45 days                                                                    |
| 317 | of sand burial_(Figure 2 B). Highest (0.057 $\pm$ 0.004 mm * cm -1 d -1 ) and lowest (0.023 $\pm$ 0.006 mm * cm -1 d -1 ) relative growth rates |
| 318 | for seedling height were shown in the T33 and T100 treatments, respectively (Figure 2B ). The pattern of change with burial                                                                                      |
| 319 | depth was similar to that described for seedling height (Figure 2 A); values were greater in the control than 100% covered by                                                                                    |
| 320 | sand (Figure 2 B; P<0.05).                                                                                                                                                                                       |
| 321 | After 45 days from initiation of the study, stem diameter ( $F_{3.56}=26.669$ , $P \leq 0.001$ ), taproot length ( $F_{3.56}=30.942$ , $P < 0.001$ ) and                                                                |
| 322 | total leaf area (F 3,56 =35.961, P<0.001) of seedlings were also affected by sand burial (Figure 3 A, B, C). Stem diameter was                                                                               |
| 323 | 20% greater (P<0.05) in the T33 than in the T0 treatment (Figure 3A). However, stem diameters were similar in the control                                                                                               |
| 324 | and T67 treatments (Figure 3A. P>0.05). Values in the T100 treatment, nevertheless, were 13.4% lower than those in the                                                                                                  |
| 325 | unburied control (Figure 3A). While taproot length was lowest in the control treatment, it was highest in the T67 treatment                                                                                             |
| 326 | (Figure 3B; 35.7% higher than that in the control; P<0.05). The total leaf area of seedlings was significantly greater in the                                                                                           |

|----------------------------|
| height growth rate         |

|---|--------------------------|

|      |                                                                                                                                                                                                                                                                                                                                                                                                                                                                                                                                                                                                                                                                                                                                                                                                                                                                                                                                                                                                                                                                                                                                                                                                                                                                                                                                                                                                                                                                                                                                                                                                                                                                                                                                                                                                                                                                                                                                                                                                                                                                                                                                    | 删除的内容: of 23.87cm 2 |
|------|------------------------------------------------------------------------------------------------------------------------------------------------------------------------------------------------------------------------------------------------------------------------------------------------------------------------------------------------------------------------------------------------------------------------------------------------------------------------------------------------------------------------------------------------------------------------------------------------------------------------------------------------------------------------------------------------------------------------------------------------------------------------------------------------------------------------------------------------------------------------------------------------------------------------------------------------------------------------------------------------------------------------------------------------------------------------------------------------------------------------------------------------------------------------------------------------------------------------------------------------------------------------------------------------------------------------------------------------------------------------------------------------------------------------------------------------------------------------------------------------------------------------------------------------------------------------------------------------------------------------------------------------------------------------------------------------------------------------------------------------------------------------------------------------------------------------------------------------------------------------------------------------------------------------------------------------------------------------------------------------------------------------------------------------------------------------------------------------------------------------------------|---------------------------------------|
| 347  | control than in the T67 and T100 treatments (Figure 3C; P<0.05). The lowest total leaf area, however, was found in the T100                                                                                                                                                                                                                                                                                                                                                                                                                                                                                                                                                                                                                                                                                                                                                                                                                                                                                                                                                                                                                                                                                                                                                                                                                                                                                                                                                                                                                                                                                                                                                                                                                                                                                                                                                                                                                                                                                                                                                                                                        |                                       |
| 3/18 | treatment (Figure 3C)                                                                                                                                                                                                                                                                                                                                                                                                                                                                                                                                                                                                                                                                                                                                                                                                                                                                                                                                                                                                                                                                                                                                                                                                                                                                                                                                                                                                                                                                                                                                                                                                                                                                                                                                                                                                                                                                                                                                                                                                                                                                                                              | 删除的内容: of 16.89cm 2 |
| 540  | incumental righte 50).                                                                                                                                                                                                                                                                                                                                                                                                                                                                                                                                                                                                                                                                                                                                                                                                                                                                                                                                                                                                                                                                                                                                                                                                                                                                                                                                                                                                                                                                                                                                                                                                                                                                                                                                                                                                                                                                                                                                                                                                                                                                                                             | 带格式的:字体:倾斜                            |
| 240  | Effects of sand burial on biomass growth and relative mass growth rate                                                                                                                                                                                                                                                                                                                                                                                                                                                                                                                                                                                                                                                                                                                                                                                                                                                                                                                                                                                                                                                                                                                                                                                                                                                                                                                                                                                                                                                                                                                                                                                                                                                                                                                                                                                                                                                                                                                                                                                                                                                             | 带格式的:字体:倾斜                            |
| 545  | pijecis of sana baria on biomass grown and readive mass, grown rate                                                                                                                                                                                                                                                                                                                                                                                                                                                                                                                                                                                                                                                                                                                                                                                                                                                                                                                                                                                                                                                                                                                                                                                                                                                                                                                                                                                                                                                                                                                                                                                                                                                                                                                                                                                                                                                                                                                                                                                                                                                                | 删除的内容: ,                       |
| 350  | There were significant differences in total seedling biomass ( $F_{3,56}=129.949$ , $P \le 0.001$ ) and its component organs[e.g., leaves                                                                                                                                                                                                                                                                                                                                                                                                                                                                                                                                                                                                                                                                                                                                                                                                                                                                                                                                                                                                                                                                                                                                                                                                                                                                                                                                                                                                                                                                                                                                                                                                                                                                                                                                                                                                                                                                                                                                                                                          | 删除的内容: respectively                   |
| 351  | $(F_{1,2}-93.965, P<0.001)$ and roots $(F_{2,2}-50.474, P=0.002)$ after the experiment. The only exception was for seedling stem                                                                                                                                                                                                                                                                                                                                                                                                                                                                                                                                                                                                                                                                                                                                                                                                                                                                                                                                                                                                                                                                                                                                                                                                                                                                                                                                                                                                                                                                                                                                                                                                                                                                                                                                                                                                                                                                                                                                                                                                   | 删除的内容: Figure 4A                      |
| 551  | $(1_{3,56}-55,505,1\times(0,001))$ and $10003$ $(1_{3,56}-50,774,1=0,002)$ after the experiment. The only exception was for second s | 删除的内容: While t                        |
| 352  | biomass (F 3,56 =2.017, P=0.122) which was similar in all sand burial treatments ( Table 1 ). Greatest total biomasses were                                                                                                                                                                                                                                                                                                                                                                                                                                                                                                                                                                                                                                                                                                                                                                                                                                                                                                                                                                                                                                                                                                                                                                                                                                                                                                                                                                                                                                                                                                                                                                                                                                                                                                                                                                                                                                                                                                                                                                                      | 删除的内容: of 272.67mg             |
| 353  | reached in the T33(369.65 $\pm$ 17.27 mg) and T67 treatments (372.50 $\pm$ 15.74 mg) (Table 1). Total biomass of seedlings was                                                                                                                                                                                                                                                                                                                                                                                                                                                                                                                                                                                                                                                                                                                                                                                                                                                                                                                                                                                                                                                                                                                                                                                                                                                                                                                                                                                                                                                                                                                                                                                                                                                                                                                                                                                                                                                                                                                                                                                                     | 删除的内容: Figure 4 A                     |
| 555  | reached in the $133,307,03\pm17,27$ mg/ and $107$ treatments $(372,30\pm13,74)$ mg/ radie 12. Total biomass of seedings was                                                                                                                                                                                                                                                                                                                                                                                                                                                                                                                                                                                                                                                                                                                                                                                                                                                                                                                                                                                                                                                                                                                                                                                                                                                                                                                                                                                                                                                                                                                                                                                                                                                                                                                                                                                                                                                                                                                                                                                                        | 删除的内容: o those shown                  |
| 354  | significantly lower in T100 treatment, than in those treatments (Table 1). Patterns shown for the biomasses of leaves and                                                                                                                                                                                                                                                                                                                                                                                                                                                                                                                                                                                                                                                                                                                                                                                                                                                                                                                                                                                                                                                                                                                                                                                                                                                                                                                                                                                                                                                                                                                                                                                                                                                                                                                                                                                                                                                                                                                                                                                                          | 删除的内容: Figure 4A                      |
| 355  | roots among treatments were similar t for the total biomass of seedlings (Table 1)                                                                                                                                                                                                                                                                                                                                                                                                                                                                                                                                                                                                                                                                                                                                                                                                                                                                                                                                                                                                                                                                                                                                                                                                                                                                                                                                                                                                                                                                                                                                                                                                                                                                                                                                                                                                                                                                                                                                                                                                                                                 | 删除的内容: of 20.9%                       |
| 555  |                                                                                                                                                                                                                                                                                                                                                                                                                                                                                                                                                                                                                                                                                                                                                                                                                                                                                                                                                                                                                                                                                                                                                                                                                                                                                                                                                                                                                                                                                                                                                                                                                                                                                                                                                                                                                                                                                                                                                                                                                                                                                                                                    | 删除的内容:                                |
| 356  | Significant differences were found in allocation of seedling biomass to leaves ( $F_{3,56}$ =12.841, P<0.001), stems ( $F_{3,56}$ =27.579,                                                                                                                                                                                                                                                                                                                                                                                                                                                                                                                                                                                                                                                                                                                                                                                                                                                                                                                                                                                                                                                                                                                                                                                                                                                                                                                                                                                                                                                                                                                                                                                                                                                                                                                                                                                                                                                                                                                                                                                         | 删除的内容: Figure 4B                      |
| 357  | $P<0.001$ ) and roots ( $F_{2.55}=7.594$ , $P<0.001$ ). On leaves, percentage biomass allocation was greatest in the T33and T67                                                                                                                                                                                                                                                                                                                                                                                                                                                                                                                                                                                                                                                                                                                                                                                                                                                                                                                                                                                                                                                                                                                                                                                                                                                                                                                                                                                                                                                                                                                                                                                                                                                                                                                                                                                                                                                                                                                                                                                                    | 删除的内容:                                |
|      |                                                                                                                                                                                                                                                                                                                                                                                                                                                                                                                                                                                                                                                                                                                                                                                                                                                                                                                                                                                                                                                                                                                                                                                                                                                                                                                                                                                                                                                                                                                                                                                                                                                                                                                                                                                                                                                                                                                                                                                                                                                                                                                                    | 删除的内容: was similar                    |
| 358  | treatment, and lowest in the control and T100 treatments (Table 2). Percentage biomass allocation to stems was greatest in                                                                                                                                                                                                                                                                                                                                                                                                                                                                                                                                                                                                                                                                                                                                                                                                                                                                                                                                                                                                                                                                                                                                                                                                                                                                                                                                                                                                                                                                                                                                                                                                                                                                                                                                                                                                                                                                                                                                                                                                         | 删除的内容: in                             |
| 359  | the T100 treatment ( $24.1 \pm 1.9\%$ ), values on stems was greater in the control than in the T33 and T67 treatments (Table 2).                                                                                                                                                                                                                                                                                                                                                                                                                                                                                                                                                                                                                                                                                                                                                                                                                                                                                                                                                                                                                                                                                                                                                                                                                                                                                                                                                                                                                                                                                                                                                                                                                                                                                                                                                                                                                                                                                                                                                                                                  | 删除的内容:,                               |
|      |                                                                                                                                                                                                                                                                                                                                                                                                                                                                                                                                                                                                                                                                                                                                                                                                                                                                                                                                                                                                                                                                                                                                                                                                                                                                                                                                                                                                                                                                                                                                                                                                                                                                                                                                                                                                                                                                                                                                                                                                                                                                                                                                    | 删除的内容:(                               |
| 360  | Finally, percentage biomass allocation to roots showed a slight decreasing trend from the control to T33 and T67 treatments                                                                                                                                                                                                                                                                                                                                                                                                                                                                                                                                                                                                                                                                                                                                                                                                                                                                                                                                                                                                                                                                                                                                                                                                                                                                                                                                                                                                                                                                                                                                                                                                                                                                                                                                                                                                                                                                                                                                                                                                        | 删除的内容: Figure 4B                      |
| 361  | (Table 2); values determined in the T100 treatment for this organ were significantly lower than in the control and T33                                                                                                                                                                                                                                                                                                                                                                                                                                                                                                                                                                                                                                                                                                                                                                                                                                                                                                                                                                                                                                                                                                                                                                                                                                                                                                                                                                                                                                                                                                                                                                                                                                                                                                                                                                                                                                                                                                                                                                                                             | 删除的内容:s                               |
|      |                                                                                                                                                                                                                                                                                                                                                                                                                                                                                                                                                                                                                                                                                                                                                                                                                                                                                                                                                                                                                                                                                                                                                                                                                                                                                                                                                                                                                                                                                                                                                                                                                                                                                                                                                                                                                                                                                                                                                                                                                                                                                                                                    | 删除的内容: Figure 4B                      |
| 362  | treatment ( Table 2 ).                                                                                                                                                                                                                                                                                                                                                                                                                                                                                                                                                                                                                                                                                                                                                                                                                                                                                                                                                                                                                                                                                                                                                                                                                                                                                                                                                                                                                                                                                                                                                                                                                                                                                                                                                                                                                                                                                                                                                                                                                                                                                                      | 删除的内容: Seeding relative growth        |
| 363  | The relative mass growth rate of seedlings was significantly affected by sand burial at the end of experiment ( $F_{3.56}$ =136.370,                                                                                                                                                                                                                                                                                                                                                                                                                                                                                                                                                                                                                                                                                                                                                                                                                                                                                                                                                                                                                                                                                                                                                                                                                                                                                                                                                                                                                                                                                                                                                                                                                                                                                                                                                                                                                                                                                                                                                                                 | 删除的内容: after                          |
|      |                                                                                                                                                                                                                                                                                                                                                                                                                                                                                                                                                                                                                                                                                                                                                                                                                                                                                                                                                                                                                                                                                                                                                                                                                                                                                                                                                                                                                                                                                                                                                                                                                                                                                                                                                                                                                                                                                                                                                                                                                                                                                                                                    | 删除的内容:=                               |
| 364  | $P \leq 0.001$ . Greatest relative mass growth rate values of $0.031 \pm 0.001$ mg $\cdot$ mg -1 day -1 were shown both in the T33 and T67                                                                                                                                                                                                                                                                                                                                                                                                                                                                                                                                                                                                                                                                                                                                                                                                                                                                                                                                                                                                                                                                                                                                                                                                                                                                                                                                                                                                                                                                                                                                                                                                                                                                                                                                                                                                                                                                                                                                                                   | 删除的内容:0                               |

Ì

| 393 | treatments (Figure 4). These values were significantly greater than those found in the control (Figure 4, $P < 0.05$ ). Lowest                            |                           | 删                 |
|-----|-----------------------------------------------------------------------------------------------------------------------------------------------------------|---------------------------|-------------------|
|     |                                                                                                                                                           |                           | 删图                |
| 394 | relative mass rates of growth were determined on seedlings grown in the T100 treatment $(0.026 \pm 0.001 \text{ mg} \cdot \text{mg}^{-1}\text{day}^{-1})$ |                           | 删图                |
| 395 | Figure 4).                                                                                                                                                | $\backslash$              | 删图                |
|     |                                                                                                                                                           | ( ) )                     | 删降                |
| 396 |                                                                                                                                                           | $\langle \rangle \rangle$ | 删                 |
| 397 | Discussion                                                                                                                                                | $\langle \rangle$         | 删                 |
| 200 |                                                                                                                                                           | $\nearrow$                | 刪                 |
| 398 | Seedling survivorship in response to permanent sand burial                                                                                                | $\langle \rangle$         | 帯
New          |
| 399 | In sand land regions, seedlings might be buried at different depths between emergence and the end of the windy season, from                               |                           | 删图                |
|     |                                                                                                                                                           |                           | 刪                 |
| 400 | late spring to early summer (Chen and Maun, 1999) and this was stimulated in our experiment. After the 45-days, partial                                   |                           | 域                 |
| 401 | burial with sand (to 33% or 67% of their height) did not influence the survival of seedling of sandy elm seedling, as there                               |                           | 帯 構
New |
| 402 | was no mortality. These results agreed with studies of He et al. (2008), Liu et al. (2008) and Qu et al. (2012), which reported                           |                           | 删                 |
|     |                                                                                                                                                           | $\langle $                | sur               |
| 403 | that survivorship, of Artemisia halodendron, Corispermum_ macrocarpum and Caragana microphylla was either maintained                                      |                           | exn               |
| 404 | or increased by moderate sand burial in Horoin Sandy Land, Survivorship of these shrubs, however, declined among plant                                    | $\backslash \backslash$   | 3.6               |
|     |                                                                                                                                                           | $\langle \ \rangle$       | 删图                |
| 405 | species once their seedlings were covered by sand either equal to or more than 100% of their height, and this was also the                                | ( )                       | 删                 |
| 406 | case for our experiment. Survival decreased sharply by a mean of 15.6%, when seedling were completely buried (to 100% of                                  | //                        | 删                 |
|     |                                                                                                                                                           | $\mathbb{N}$              | 刪                 |
| 407 | their height), while deeper sand burial resulted in no survival of sandy elm at all, as seedlings withered and rotted in the                              |                           | 刪                 |
| 408 | soil .This precise threshold offers a clear primary explanation for the absence of sandy elm seedling after relatively deep                               |                           | 删图                |
|     |                                                                                                                                                           | ```                       | 刪                 |
| 409 | sand burial. Field survey in recent years has showed that serious land degradation and reduction of vegetation cover has                                  |                           |                   |
| 410 | aggravated sand mobility, particularly in the leeward and semi-fixed dunes. Seedlings could successfully complete their                                   |                           |                   |

| 余的内容: 5                    |
|-----------------------------------|
| 余的内容: 5                    |
| 余的内容: as                          |
| 余的内容:,                            |
| 余的内容: just                 |
| 余的内容:                             |
| 余的内容: (                    |
| 余的内容: 5                    |
| 各式的: 字体:(默认)Times
Roman |
| 余的内容: continual                   |
| 余的内容: by                   |
| 代码已更改                             |
| 各式的: 字体:(默认)Times
Roman |
|                                   |

| 1除的内容: -experiment,             |  |  |  |
|----------------------------------------|--|--|--|
| urvivorship remained unchanged         |  |  |  |
| hen seedlings were either unburied or  |  |  |  |
| xposed to partial [i.e., 1.8 (T33) and |  |  |  |
| .6 (T67) cm soil depth].               |  |  |  |
| ]除的内容: by                       |  |  |  |
| ]除的内容: vigor                    |  |  |  |
| ]除的内容: varied                   |  |  |  |
| 引除的内容: height                   |  |  |  |
| ]除的内容: Whilst the values        |  |  |  |
| 引除的内容: significantly            |  |  |  |
| 小除的内容: by a mean of             |  |  |  |

| 433 | periodic recruitments only taking advantage of scarce favorable spatio-temporal chances (Tian et al., 2015; Wu et al., 2013)       |                                                                                                                                                                                                                                                                                                                                                                                                                                                                                                                                                                                                                                                                                                                                                                                                                                                                                                                                                                                                                                                                                                                                                                                                                                                                                                                                                                                                                                                                                                                                                                        |
|-----|------------------------------------------------------------------------------------------------------------------------------------|------------------------------------------------------------------------------------------------------------------------------------------------------------------------------------------------------------------------------------------------------------------------------------------------------------------------------------------------------------------------------------------------------------------------------------------------------------------------------------------------------------------------------------------------------------------------------------------------------------------------------------------------------------------------------------------------------------------------------------------------------------------------------------------------------------------------------------------------------------------------------------------------------------------------------------------------------------------------------------------------------------------------------------------------------------------------------------------------------------------------------------------------------------------------------------------------------------------------------------------------------------------------------------------------------------------------------------------------------------------------------------------------------------------------------------------------------------------------------------------------------------------------------------------------------------------------|
| 434 | Maun (1981, 1997) and Disraeli (1984) also indicated that a certain tolerance to partial sand burial was an effective strategy     |                                                                                                                                                                                                                                                                                                                                                                                                                                                                                                                                                                                                                                                                                                                                                                                                                                                                                                                                                                                                                                                                                                                                                                                                                                                                                                                                                                                                                                                                                                                                                                        |
|     |                                                                                                                                    | $\left[ \right]$                                                                                                                                                                                                                                                                                                                                                                                                                                                                                                                                                                                                                                                                                                                                                                                                                                                                                                                                                                                                                                                                                                                                                                                                                                                                                                                                                                                                                                                                                                                                                       |
| 435 | for survival and subsequent establishment of seedlings in sandy environments. Most seedlings of the grass Distichlis spicata       |                                                                                                                                                                                                                                                                                                                                                                                                                                                                                                                                                                                                                                                                                                                                                                                                                                                                                                                                                                                                                                                                                                                                                                                                                                                                                                                                                                                                                                                                                                                                                                        |
| 436 | died when completely covered by sand in North America (Brown, 1997; Li et al., 2015), while some Artemisia squarrosum              |                                                                                                                                                                                                                                                                                                                                                                                                                                                                                                                                                                                                                                                                                                                                                                                                                                                                                                                                                                                                                                                                                                                                                                                                                                                                                                                                                                                                                                                                                                                                                                        |
| 437 | seedlings still remained alive even though sand burial depths reached 266% of the initial seedling height in the Horqin Sandy      | and the second se                                                                                                                                                                                                                                                                                                                                                                                                                                                                                                                                                                                                                                                                                                                                                                                                                                                                    |
| 438 | Land (Li et al. 2015). Thus, compared with other species, seedling of sandy elm sowed moderate resistance to sand burial.          | AN AVAILABLE AND AVAILABLE |
| 439 | Harris and Davy (1987) and Perumal and Maun (2006) suggested that plant energy exhaustion and suppression of                       |                                                                                                                                                                                                                                                                                                                                                                                                                                                                                                                                                                                                                                                                                                                                                                                                                                                                                                                                                                                                                                                                                                                                                                                                                                                                                                                                                                                                                                                                                                                                                                        |
| 440 | photosynthesis were implicated in the severely reduced intense radiation and high temperature to some extent. Seedlings of         |                                                                                                                                                                                                                                                                                                                                                                                                                                                                                                                                                                                                                                                                                                                                                                                                                                                                                                                                                                                                                                                                                                                                                                                                                                                                                                                                                                                                                                                                                                                                                                        |
| 441 | sandy elm have adapted to extreme conditions and previous research has confirmed that sandy elm had a higher transpiration         |                                                                                                                                                                                                                                                                                                                                                                                                                                                                                                                                                                                                                                                                                                                                                                                                                                                                                                                                                                                                                                                                                                                                                                                                                                                                                                                                                                                                                                                                                                                                                                        |
| 442 | rate and stomatal conductance with lower photosynthesis water-use efficiency and less sensitivity to high temperature and          |                                                                                                                                                                                                                                                                                                                                                                                                                                                                                                                                                                                                                                                                                                                                                                                                                                                                                                                                                                                                                                                                                                                                                                                                                                                                                                                                                                                                                                                                                                                                                                        |
| 443 | irradiance, compared with other native tree species such as Malus baccata, Prunus padus and Pinus sylvestris, especially in        |                                                                                                                                                                                                                                                                                                                                                                                                                                                                                                                                                                                                                                                                                                                                                                                                                                                                                                                                                                                                                                                                                                                                                                                                                                                                                                                                                                                                                                                                                                                                                                        |
| 444 | the midday (Park et al., 2012).                                                                                                    |                                                                                                                                                                                                                                                                                                                                                                                                                                                                                                                                                                                                                                                                                                                                                                                                                                                                                                                                                                                                                                                                                                                                                                                                                                                                                                                                                                                                                                                                                                                                                                        |
| 445 | Effects of sand burial on seedling morphological traits                                                                            |                                                                                                                                                                                                                                                                                                                                                                                                                                                                                                                                                                                                                                                                                                                                                                                                                                                                                                                                                                                                                                                                                                                                                                                                                                                                                                                                                                                                                                                                                                                                                                        |
| 446 | Sand burial , modifies, the environment of living plants, forming new microhabitat available for seedling (Disraeli, 1984;         |                                                                                                                                                                                                                                                                                                                                                                                                                                                                                                                                                                                                                                                                                                                                                                                                                                                                                                                                                                                                                                                                                                                                                                                                                                                                                                                                                                                                                                                                                                                                                                        |
| 447 | Sun et al., 2014). Plants would be expected to adjust their morphological performances and developments to maximize                |                                                                                                                                                                                                                                                                                                                                                                                                                                                                                                                                                                                                                                                                                                                                                                                                                                                                                                                                                                                                                                                                                                                                                                                                                                                                                                                                                                                                                                                                                                                                                                        |
| 448 | photosynthetic efficiency and sustain survival (Wang et al., 2014). Our results demonstrated that various seedling                 |                                                                                                                                                                                                                                                                                                                                                                                                                                                                                                                                                                                                                                                                                                                                                                                                                                                                                                                                                                                                                                                                                                                                                                                                                                                                                                                                                                                                                                                                                                                                                                        |
| 449 | morphological traits (i.e., height; stem diameter; taproot length; total leaf area; dry biomass; partitioning of biomass to shoots |                                                                                                                                                                                                                                                                                                                                                                                                                                                                                                                                                                                                                                                                                                                                                                                                                                                                                                                                                                                                                                                                                                                                                                                                                                                                                                                                                                                                                                                                                                                                                                        |
| 450 | and roots; RHGR and RMGR ) were increased by partial butial, especially at T33. Thus seedling height of sandy elm was              |                                                                                                                                                                                                                                                                                                                                                                                                                                                                                                                                                                                                                                                                                                                                                                                                                                                                                                                                                                                                                                                                                                                                                                                                                                                                                                                                                                                                                                                                                                                                                                        |

|-----------------------------------------|---|--|--|
| covered the whole, original seedling    |   |  |  |
| height [i.e., 5.4cm soil depth (T100)]. | J |  |  |
| exceeded 33% of the original seedling   |   |  |  |
| height [i.e., 7.2cm soil depth (T133)], |   |  |  |
| there was no seedling survival, only    |   |  |  |
| withered and rotted.(Tian et al., 2015; |   |  |  |
| Wu et al., 2013)Harris and Davy (       | J |  |  |
| 1 B 40                                  | ٦ |  |  |

| 494 | greater in the partially buried than in the unburied and completely buried treatments, which indicated, that partial sand burial    |
|-----|-------------------------------------------------------------------------------------------------------------------------------------|
| 495 | stimulated stem elongation. This might be explained by the fcat that the processes of growth and elongation, benefiting from |
| 496 | improved, water maintenance and nutrient uptake in these arid and semi-arid regions (Li et al., 2015). In Horqin Sandy Land,        |
| 497 | dry sandy layer was reported in the first 5 cm from soil surface level in semi-fixed dunes. This suggests that suitable sand        |
| 498 | burial depth could be beneficial in reducing soil temperature and keep moisture for root uptake, which are critical to              |
| 499 | seedlings survival and resource capturing (Niu et al., 2015). Our finding was, also consistent with previous research reported      |
| 500 | from Disraeli (1984), who reported that partial burial stimulated growth of Ammophila breviligulata in coastal dunes of             |
| 501 | northeastern North America, Belcher (1977) also determind that seedling heights of Rosa rugosa was higher in the partial            |
| 502 | than in the unburied and completely continuous sand burial treatments in the desert.                                  |
| 503 | Seedling height growth rate was a critical parameter to determine the speed of growth. The greater RHGR of seedlings in the         |
| 504 | partial (T33 and T67) than in the unburied and completely buryied, treatments (T100) was an indication that partial burial did      |
| 505 | contribute to a greater seedling height after a 45-day-growth period via accelerating the speed of growth in height.         |
| 506 | Nevertheless, the greatest seedling growth in height in the T33 treatment came from its greatest RHGR in this than any other        |
| 507 | treatments. Liu et al. (2008) and Miao et al. (2012) also found that shallow soil burial depths could promote the relative          |
| 508 | growth rates in height on Salix gordejevii, Artemisia wudanica and A rtemisia halodendron. However, there is not universally |
| 509 | true as some species(e.g., Artemisi gmelinii) have decreased their growth rates as a result of sand burial (Liu et al., 2008).      |
| 510 | These findings confirmed that the phenotypic response to the degree of sand burial might be species-specific.                       |
| 511 | Compared with the unburied treatment, partial burial treatments fostered increments in stem diameter and taproot length. Sun        |

|--------------------------------|
|                                |

| 553 | et al. (2014a) also showed that seedling diameter and taproot length of Suaeda salsa were increased by a partial burial          |
|-----|----------------------------------------------------------------------------------------------------------------------------------|
| 554 | treatment in the coastal marches of the Yellow River estuary, China. Caldwell et al. (1998) found that increases in taproot      |
| 555 | length were conducive to a greater nutrient and water uptake from deeper soil depths. Total leaf area, however, was either       |
| 556 | similar or lower in the partial burial than in the unburied treatment. This was in agreement with the results of Liu and Guo     |
| 557 | (2005), who noted that increasing depth of sand burial decreased the total leaf area of Caragana intermedia.                     |
| 558 | Changes of biomass allocation and relative mass growth rate, which are involved in successful seedling recruitment, have         |
| 559 | been regarded as the results of variations in plant adaptive strategies in response to environmental changes, (Wu et al., 2013). |
| 560 | Appropriate resource allocation is essential for plant establishment and growth (Bazzaz, 1997). Also, plants may shift           |
| 561 | resource allocation to minimize the effects of external environmental changes (Maun, 1997; Ni et al., 2015). Numerous            |
| 562 | findings, especially those on sandy environments, have reported that plants could withstand episodes sand burial by changing     |
| 563 | biomass allocation. Some species (e.g. Artemisia ordosia, Elymusfarctus) may transfer biomass from underground to leaf and       |
| 564 | stem organs (Brown, 1997; Li et al., 2010), while others (e.g., Caragana_microphylla, Nitraria_sphaerocarpa) have either         |
| 565 | maintained or increased biomass allocation to roots_(He et al., 2008; Sykes and Wilson, 1990). In our experiment, it was         |
| 566 | somewhat surprising that no differences were, observed in the dry biomass of seedling stems among all sand burial treatments.    |
| 567 | However, seedlings showed an increased stem diameter in the T33 sand burial treatment; similar to results of Zhao et             |
| 568 | al.(2015), this was most likely because of fresh stem had a greater water content in the T33 than in the other treatments.       |
| 569 | Additionally, partial burial treatments produced, greater dry biomass for leaves, roots and the whole seedlings in comparison    |
| 570 | with unburied and completely sand burial treatments. Previous studies also determined that 67% burial of seedlings of the        |

|----------------------------------------|
| nutrients obtained from deep depths by |
| dicots species could be released at    |
| shallower soil depths if soil is drier |
| than plant tissues at those depths     |
| —————————————————————————————————————  |

| 611 | shrub Caragana intermedia determined a greater biomass allocation to leaves and stems than to roots, compared with values       |
|-----|---------------------------------------------------------------------------------------------------------------------------------|
| 612 | in the unburied control (Xu et al., 2013). Meanwhile, there was a trend for an increasing aboveground (leaf + stem), and a      |
| 613 | decreasing belowground allocation with increasing burial depth for sandy elm seedlings. Nearly 50% of the total seedling        |
| 614 | biomass corresponded to leaves. This indicated sufficient leaves are necessary on sandy elm to sustain photosynthesis and       |
| 615 | maintain evapotranspiration after exposure to high temperature and intense irradiance environments during the growing           |
| 616 | season (Dulamsuren et al., 2009; Li et al., 2003; Park et al., 2012). The relatively investment in root was slight decreased,   |
| 617 | which indicated that, on one hand, greater soil moisture availability weakened the dependence on root function, and on the      |
| 618 | other hand, the plant's need to divert biomass aboveground for light interception and net assimilation rate (Maun, 1994; Sun    |
| 619 | et al., 2010; Wang et al., 2014)                                                                                                |
| 620 | Relative mass growth rates measure the mean efficiency rate for producing new biomass (Walck et al., 1999). Dalling and         |
| 621 | Hubbell (2002) showed, that seedling growth rate, was a better determinant of successful seedling establishment than biomass.   |
| 622 | In our experiment, relative mass growth rates were higher in the partial than in the other treatments, indicating that moderate |
| 623 | sand burials were beneficial for a rapid biomass increase. However, all mass relative growth rates were small compared with     |
| 624 | those of other plant species (e.g., Artemisia wudanica, Solidago shortii and Solidago. altissima) in the same area. This,       |
| 625 | suggests that the relative lower biomass accumulation on sandy elm seedlings during the first growing season places these       |
| 626 | seedlings at a disadvantage when considering soil resource competition and coexistence with other species (Brown, 1997;         |
| 627 | Liu et al., 2014; Wu et al., 2013). Reduced dry matter accumulation could also have contributed to increased mortality.         |
| 628 | Although there were striking effects on hiomass accumulation and allocation reflecting the plasticity of various                |

New Roman | 字体: | (默认) | Times |
|---------------------------|-----|------|-------|
| 域代码已更                     | 改   |      |       |
New Roman | 字体: | (默认) | Times |
New Roman | 字体: | (默认) | Times |
| 域代码已更                     | 改   |      |       |
New Roman | 字体: | (默认) | Times |
New Roman | 字体: | (默认) | Times |
New Roman | 字体: | (默认) | Times |
New Roman | 字体: | (默认) | Times |

|---|---------------------------|------|------|-------|--|--|
New Roman | 字体:  | (默认) | Times |  |  |
New Roman | 字体:  | (默认) | Times |  |  |
New Roman | 字体:  | (默认) | Times |  |  |

| 658 | morphological traits, Partial sand burial treatments did not change the survivorship, and these treatments facilitate individual   |
|-----|------------------------------------------------------------------------------------------------------------------------------------|
| 659 | seedling growth and population regeneration through phenotylic plasticity or various morphological traits. Our study,              |
| 660 | however, was conducted in pots. We have to recognize it has limitations in comparison to field studies. Under natural              |
| 661 | rangeland conditions, some factors (e.g., abrasion of plant tissues by sand grains; grazing by herbivores and granivores)          |
| 662 | might reduce or eliminate some of the positive effects of the partial burial treatments (Dulamsuren et al., 2009; Jeffreyt et al., |
| 663 | 2009). Furthermore, we found evidence of allometry of each plant proportion with increasing seedling age in our experiment.        |
| 664 | Therefore, more comprehensive studies on physiological and biochemical mechanisms involved on sandy elm seedling                   |
| 665 | survivorship and performance under field, sand burial conditions at different growth stage in future research.                     |
| 666 | Conclusion                                                                                                                         |
| 667 | Sand burial affected seedling survivorship, growth and biomass allocation of Ulmus pumila var. sabulosa through phenotypic         |
| 668 | plasticity of morphological traits. Seedlings of sandy elm showed adaptive responses to moderate sand burial, consistent with      |
| 669 | its evolution in the sandy environment. Partial sand burial treatment did not influence seedling survivorship, but complete        |
| 670 | sand burial significantly increased mortality. Compared with the unburied treatment, seedling height, relative height growth       |
| 671 | rates, taproot length, total biomass and relative mass growth rates were stimulated by partial burial with sand. At the same       |
| 672 | time, percentage biomass allocation of seedlings was changed, diverting, more biomass to aboveground organs (e.g., leaf and        |
| 673 | stem) to sustain normal photosynthesis and evapo-transpiration. Complete sand burial after seedling emergence, however,     |
| 674 | inhibited their growth, and even resulted in seedling death. Consequently, burial depths should be controlled by making            |
| 675 | enclosures and increasing vegetation coverage to facilitate regeneration or re-establishment of sandy elm, The observed            |

|--------------------------------|--|

( . . .

• • • •

environments **删除的内容:** H

| 711        | variation in all parameters has defined the tolerance of Ulmus pumila var. sabulosa to sandy environments and its capacity to                                     |                      | 删除的内容: indicated                                     |        |
|------------|-------------------------------------------------------------------------------------------------------------------------------------------------------------------|----------------------|------------------------------------------------------|--------|
| 712        | acclimate them. Hence our research provides a theoretical support for recruitment in sandy sparse elm woodlands.                                                  |                      | 删除的内容: could tolerate partia sand burial, and | al     |
| 714        | Acknowledgements                                                                                                                                                  |                      | 删除的内容: that it could                                 |        |
| 715        | This study was supported by National Natural Science Foundation of China (31370706) and the Wulanaodu Desertification                                             |                      | 删除的内容: e                                             |        |
| 716        | Experimental Station of the Institute of Applied Ecology, Chinese Academy of Sciences. We thank Yongming Luo ,Xuehua                                              | /                    | 删除的内容: to sandy environme                            | ents   |
| 717        | Li, Quantai Zhou, Hongmei wang, Melyu Jia, Ya Liu and Xu Han for assistance during the experiment. Thanks Authony J.                                       | $\langle \rangle$    |                                                      | AILS . |
| /18        | Davy from University of East Anglia for guiding of statistical analysis and language polishing for the original manuscript,                                       |                      | 删除的内容: U'                                     |        |
| 720
721 |                                                                                                                                                                   |                      |                                                      |        |
| 722        | References                                                                                                                                                        |                      |                                                      |        |
| 723        | Bazzaz, F. A.: Allocation of Resources in Plants: State of the Science and Critical Questions. In: Plant Resource Allocation,                                     |                      | 带格式的: 缩进:左侧: 0 厘                              | ī米     |
| 724        | Academic Press, San Diego, 1997.                                                                                                                                  |                      |                                                      |        |
| 725        | Belcher, C. R.: Effect of sand cover on the survival and vigor of Rosa rugosa Thunb, Int J Biometeorol, 21, 276-280, 1977.                                        |                      |                                                      |        |
| 726        | Bellamy, P. H., Loveland, P. J., Bradley, R. I., Lark, R. M., and Kirk, G. J. D.: Carbon losses from all soils across England                                     |                      |                                                      |        |
| 727        | and Wales 1978-2003, Nature, 437, 245-248, 2005.                                                                                                                  |                      |                                                      |        |
| 728
729 | Berendse, F., van Ruijven, J., Jongejans, E., and Keesstra, S.: Loss of Plant Species Diversity Reduces Soil Erosion Resistance, Ecosystems, 18, 881-888, 2015.   |                      |                                                      |        |
| 730
731 | Brevik, E. C., Cerd à A., Mataix-Solera, J., Pereg, L., Quinton, J. N., Six, J., and Van Oost, K.: The interdisciplinary nature of SOIL, SOIL, 1, 117-129, 2015.  |                      |                                                      |        |
| 732
733 | Brown, J. F.: Effects of Experimental Burial on Survival, Growth, and Resource Allocation of Three Species of Dune Plants, Journal of Ecology, 85, 151-158, 1997. |                      |                                                      |        |
| 734        | Cao, C.Y., Jiang, D.M., Teng, X.H., Jiang, Y., Liang, W.J., and Cui, Z.B.: Soil chemical and microbiological properties along                                     |                      |                                                      |        |
| 735
736 | a chronosequence of Caragana microphylla Lam. plantations in the Horqin sandy land of Northeast China, Applied Soil Ecology, 40, 78-85, 2008.              |                      |                                                      |        |
| 737        | Cao, C.Y., Jiang, S.Y., Zhang, Y., Zhang, F.X., and Han, X.S.: Spatial variability of soil nutrients and microbiological                                          |                      |                                                      |        |
| 738        | properties after the establishment of leguminous shrub Caragana microphylla Lam. plantation on sand dune in the                                            |                      |                                                      |        |
| 739        | Horqin Sandy Land of Northeast China, Ecological Engineering, 37, 1467-1475, 2011.                                                                                |                      |                                                      |        |
| 740        | Cerd à A.: The influence of aspect and vegetation on seasonal changes in erosion under rainfall simulation on a clay soil in                                      |                      |                                                      |        |
| 741        | Spain, Canadian Journal of Soil Science, 78, 321-330, 1998.                                                                                                       |                      |                                                      |        |
| 742
743 | Chen, H. and Maun, M. A.: Effects of sand burial depth on seed germination and seedling emergence of Cirsium pitcheri , Plant Ecology, 140, 53-60, 1999.   |                      |                                                      |        |

| 753
754 | Cheplick, G. P. and Grandstaff, K.: Effects of sand burial on purple sandgrass ( Triplasis purpurea ): the significance of seed heteromorphism, Plant Ecology, 133, 79-89, 1997. |             |
|------------|-----------------------------------------------------------------------------------------------------------------------------------------------------------------------------------------|-------------|
| 755        | Disraeli, D. J.: The Effect of Sand Deposits on the Growth and Morphology of Ammophila Breviligulata, Journal of Ecology,                                                               | 带格式的:字体:非倾斜 |
| 756        | 72, 145-154, 1984.                                                                                                                                                                      |             |
| 757        | Dulamsuren, C., Hauck, M., Nyambayar, S., Bader, M., Osokhjargal, D., Oyungerel, S., and Leuschner, C.: Performance of                                                                  |             |
| 758        | Siberian elm (Ulmus pumila) on steppe slopes of the northern Mongolian mountain taiga: Drought stress and herbivory                                                                     |             |
| 759        | in mature trees, Environmental and Experimental Botany, 66, 18-24, 2009.                                                                                                                |             |
| 760        | Foley, J. A., DeFries, R., Asner, G. P., Barford, C., Bonan, G., Carpenter, S. R., Chapin, F. S., Coe, M. T., Daily, G. C., Gibbs,                                                      |             |
| 761        | H. K., Helkowski, J. H., Holloway, T., Howard, E. A., Kucharik, C. J., Monfreda, C., Patz, J. A., Prentice, I. C.,                                                                      |             |
| 762        | Ramankutty, N., and Snyder, P. K.: Global Consequences of Land Use, Science, 309, 570-574, 2005.                                                                                        |             |
| 763        | Gabarr ón-Galeote, M. A., Mart nez-Murillo, J. F., Quesada, M. A., and Ruiz-Sinoga, J. D.: Seasonal changes in the soil                                                                 |             |
| 764        | hydrological and erosive response depending on aspect, vegetation type and soil water repellency in different                                                                           |             |
| 765        | Mediterranean microenvironments, Solid Earth, 4, 497-509, 2013.                                                                                                                         |             |
| 766        | Harris, D. and Davy, A. J.: Seedling Growth in Elymus farctus after Episodes of Burial with Sand, Annals of botany, 60,                                                                 | 带格式的:字体:非倾斜 |
| 767        | 587-593, 1987.                                                                                                                                                                          |             |
| 768        | He, Y.H., Zhao, H.L., Zhao, X.Y., and Liu, X.P.: Effects of different sand burial depths on growth and biomass allocation in                                                            |             |
| 769        | Caragana microphylla seedlings, Arid Land Geography, 31, 701-706, 2008.                                                                                                                 |             |
| 770        | Jeffreyt, B., Bobbie, M. M., Johnj, B., Dennisc, G., Robertj, L., and Jhonathane, E.: Sand Abrasion Injury and Biomass                                                                  |             |
| 771        | Partitioning in Cotton Seedlings, Agronomy Journal, 101, 1297-1303, 2009.                                                                                                               |             |
| 772        | Jiang, D.M., Liu, Z.M., Cao, C.Y., Kou, Z.W., and Wang, R.Y.: Desertification and Ecological Restoration of Keerqin Sandy                                                               |             |
| 773        | Land, China Environmental Science Press, Beijing, 2003.                                                                                                                                 |             |
| 774        | Jiang, D.M., Tang, Y., and Busso, C. A.: Effects of vegetation cover on recruitment of Ulmus pumila L. in Horqin Sandy                                                                  |             |
| 775        | Land, northeastern China, Journal of Arid Land, 6, 343-351, 2013.                                                                                                                       |             |
| 776        | Li, J., Qu, H., Zhao, H.L., Zhou, R.L., Yun, J.Y., and Pan, C.C.: Growth and physiological responses of Agriophyllum                                                                    |             |
| 777        | squarrosum to sand burial stress, Journal of Arid Land, 7, 94-100, 2015.                                                                                                                |             |
| 778        | Li, S.L., Werger, M. A., Zuidema, P., Yu, FH., and Dong, M.: Seedlings of the semi-shrub Artemisia ordosica are resistant                                                               |             |
| 779        | to moderate wind denudation and sand burial in Mu Us sandland, China, Trees, 24, 515-521, 2010.                                                                                         |             |
| 780        | Li, X.H., Jiang, D.M., Zhou, Q.L., and Oshida, T.: Soil seed bank characteristics beneath an age sequence of Caragana                                                                   |             |
| 781        | microphylla shrubs in the Horqin Sandy Land region of northeastern China, Land Degradation & Development, 25,                                                                           |             |
| 782        | 236-243, 2014.                                                                                                                                                                          |             |
| 783        | Li, Y. G., Jiang, G. M., Niu, S. L., Liu, M. Z., Peng, Y., Yu, S. L., and Gao, L. M.: Gas Exchange and Water Use Efficiency                                                             |             |
| 784        | of Three Native Tree Species in Hunshandak Sandland of China, Photosynthetica, 41, 227-232, 2003.                                                                                       |             |
| 785        | Liu, B., Liu, Z.M., and Guan, D.X.: Seedling growth variation in response to sand burial in four Artemisia species from                                                                 |             |
| 786        | different habitats in the semi-arid dune field, Trees, 22, 41-47, 2008.                                                                                                                 |             |
| 787        | Liu, B., Liu, Z.M., Lü, X.T., Maestre, F. T., and Wang, L.X.: Sand burial compensates for the negative effects of erosion on                                                            |             |
| 788        | the dune-building shrub Artemisia wudanica, Plant and Soil, 374, 263-273, 2014.                                                                                                         |             |
| 789        | Liu, H. and Guo, K.: The impacts of sand burial on seedling development of Caragana intermedia, Acta Ecologica Sinica, 25,                                                              | 带格式的:字体:非倾斜 |

- 790 2550-2555, 2005.
- Liu, Z.M., Zhu, J.L., and Deng, X.: Arrival vs. retention of seeds in bare patches in the semi-arid desertified grassland of
   Inner Mongolia, northeastern China, Ecological Engineering, 49, 153-159, 2012.
- 793 Ma, C. G.: A Provenance Test of White Elm (Ulmus-Pumila L) in China, Silvae Genet, 38, 37-44, 1989.
- Maun, M. A.: Adaptations enhancing survival and establishment of seedlings on coastal dune systems, Vegetatio, 111, 59-70,
   1994.
- Maun, M. A.: Adaptations of plants to burial in coastal sand dunes, Canadian Journal of Botany-Revue Canadienne De
   Botanique, 76, 713-738, 1997.
- Maun, M. A. and Lapierre, J.: Effects of Burial by Sand on Seed Germination and Seedling Emergence of Four Dune Species,
   American Journal of Botany, 73, 450-455, 1986.
- Miao, R.H., Jiang, D.M., Musa, A., Zhou, Q.L., Guo, M.X., and Wang, Y.C.: Effectiveness of shrub planting and grazing
   exclusion on degraded sandy grassland restoration in Horqin sandy land in Inner Mongolia, Ecological Engineering, 74,
   164-173, 2014.
- Miner, B. G., Sultan, S. E., Morgan, S. G., Padilla, D. K., and Relyea, R. A.: Ecological consequences of phenotypic
   plasticity, Trends in Ecology & Evolution, 20, 685-692, 2005.
- Ni, J., Luo, D. H., Xia, J., Zhang, Z. H., and Hu, G.: Vegetation in karst terrain of southwestern China allocates more
   biomass to roots, Solid Earth, 6, 799-810, 2015.
- Niu, C. Y., Musa, A., and Liu, Y.: Analysis of soil moisture condition under different land uses in the arid region of Horqin
   sandy land, northern China, Solid Earth, 6, 1157-1167, 2015.
- Park, Y. D., Lee, D. K., Batkhuu, N. O., Tsogtbaatar, J., Combalicer, M. S., Go, E., Park, and Su, Y. W.: Woody Species
  Variations in Biomass Allocation, Photosynthetic WUE and Carbon Isotope Composition under Natural Drought
  Condition in Mongolia, Journal of Environmental Science & Management, 2012. 29-37, 2012.
- Perumal, V. J. and Maun, M. A.: Ecophysiological response of dune species to experimental burial under field and controlled
   conditions, Plant Ecology, 184, 89-104, 2006.
- Qian, J.Q., Liu, Z.M., Hatier, J. H. B., and Liu, B.: The Vertical Distribution of Soil Seed Bank and Its Restoration
  Implication in an Active Sand Dune of Northeastern Inner Mongolia, China, Land Degradation & Development, 2015.
  2015.
- 817 Qiu, S.W.: Study on the formation and evolution of Horqin Sandy Land, Scientia Geographica Sinica, 9, 317-328, 1989.
- Qu, H., Zhao, H.L., Zhou, R.L., Zuo, X.A., Luo, Y., Wang, J., and Barron, J. O.: Effects of sand burial on the survival and
   physiology of three psammophytes of Northern China, African Journal of Biotechnology, 11, 4518-4529, 2012.
- 820 Schl ütz, F., Dulamsuren, C., Wieckowska, M., Mühlenberg, M., and Hauck, M.: Late Holocene vegetation history suggests
- 821 natural origin of steppes in the northern Mongolian mountain taiga, Palaeogeography, Palaeoclimatology, Palaeoecology,
   822 261, 203-217, 2008.
- Shi, L., Zhang, Z. J., Zhang, C. Y., and Zhang, J. Z.: Effects of sand burial on survival, growth, gas exchange and biomass
   allocation of *Ulmus pumila* seedlings in the Hunshandak Sandland, China, Annals of botany, 94, 553-560, 2004.
- 825 Smith, P., Cotrufo, M. F., Rumpel, C., Paustian, K., Kuikman, P. J., Elliott, J. A., McDowell, R., Griffiths, R. I., Asakawa, S.,
- 826 Bustamante, M., House, J. I., Sobock á J., Harper, R., Pan, G., West, P. C., Gerber, J. S., Clark, J. M., Adhya, T., Scholes,

- R. J., and Scholes, M. C.: Biogeochemical cycles and biodiversity as key drivers of ecosystem services provided by soils,
   SOIL, 1, 665-685, 2015.
- Sun, Z.G., Mou, XJ., Lin, G.H., Wang, L.L., Song, H.L., and Jiang, H.: Effects of sediment burial disturbance on seedling
   survival and growth of *Suaeda salsa* in the tidal wetland of the Yellow River estuary, Plant and Soil, 337, 457-468, 2010.
- 831 Sun, Z.G., Song, H.L., Sun, W.G., and Sun, J.K.: Effects of continual burial by sediment on morphological traits and dry
- mass allocation of *Suaeda salsa* seedlings in the Yellow River estuary: An experimental study, Ecological Engineering,
   68, 176-183, 2014.
- Sykes, M. T. and Wilson, J. B.: An experimental investigation into the response of New Zealand sand dune species to
   different depths of burial by sand, Acta Botanica Neerlandica, 39, 171-181, 1990.
- Tang, J., Busso, C., Jiang, D.M., Wang, Y.C., Wu, D.F., Musa, A., Miao, R.H., and Miao, C.P.: Seed Burial Depth and Soil
  Water Content Affect Seedling Emergence and Growth of *Ulmus pumila* var. sabulosa in the Horqin Sandy Land,
  Sustainability, 8, 68, 2016.
- Tang, J., Jiang, D.M., and Wang, Y.C.: A review on the process of seed-seedling regeneration of *Ulmus pumila* in sparse
   forest grassland, Chinese Journal of Ecology, 33, 1114-1120, 2014.
- Tang, Y., Jiang, D.M., and Lü, X.T.: Effects of Exclosure Management on Elm (*Ulmus Pumila*) Recruitment in Horqin Sandy
   Land, Northeastern China, Arid Land Research and Management, 28, 109-117, 2013.
- Tian, F.P., Liu, Y., Wu, G.L., and Shi, S.L.: Seedling recruitment of forb species under experimental microhabitats in alpine
   grassland, Pakistan Journal of Botany, 47, 2127-2134, 2015.
- Verheijen, F. G. A., Jones, R. J. A., Rickson, R. J., and Smith, C. J.: Tolerable versus actual soil erosion rates in Europe,
  Earth-Science Reviews, 94, 23-38, 2009.
- Walck, J. L., Baskin, J. M., and Baskin, C. C.: Relative competitive abilities and growth characteristics of a narrowly
  endemic and a geographically widespread Solidago species (Asteraceae), American Journal of Botany, 86, 820-828,
  1999.
- Wang, D., Zhu, Y. J., Wu, G. L., and Feng, J.: Seedling performance within eight different seed-size alpine forbs under
   experimentation with irradiance and nutrient gradients, Pakistan Journal of Botany, 46, 1261-1268, 2014.
- Wang, J., Zhou, R.L., Zhao, H.L., Zhao, Y., and Hou, Y.: Growth and physiological adaptation of *Messerschmidia sibirica* to
   sand burial on coastal sandy, Acta Ecologica Sinica, 32, 4291-4299, 2012.
- Wang, K.B., Deng, L., Ren, Z.P., Li, J.P., and Shangguan, Z.: Grazing exclusion significantly improves grassland ecosystem
   C and N pools in a desert steppe of Northwest China, Catena, 137, 441-448, 2016.
- Wu, G. L., Feng, J., Shi, Z. H., and Du, G. Z.: Plant seedling performance traits impact on successful recruitment in various
   microhabitats for five Alpine Saussurea species, Pakistan Journal of Botany, 45, 61-71, 2013.
- Xu, L., Huber, H., During, H. J., Dong, M., and Anten, N. P. R.: Intraspecific variation of a desert shrub species in phenotypic plasticity in response to sand burial, New Phytologist, 199, 991-1000, 2013.
- Yan, Q.L., Liu, Z.M., Zhu, J.J., Luo, Y.M., Wang, H.M., and Jiang, D.: Structure, Pattern and Mechanisms of Formation of
   Seed Banks in Sand Dune Systems in Northeastern Inner Mongolia, China, Plant & Soil, 277, 175-184, 2005.
- Yang, H.L., Cao, Z.P., Dong, M., Ye, Y.Z., and Huang, z.: Effects of sand burying on caryopsis germination and seedling
   growth of *Bromus inermis* Leyss, Chinese Journal of Applied Ecology, 18, 2438-2443, 2007.

[revised manuscript text omitted]

Yang, L.M., Zhou, G.S., Wang, G.H., and Wang, Y.H.: Effect of human activities on soil environment and plant species

| 901        | Table legends                                                                                                                                                                                                                                                                                                                                                                                                                                                                                                                                                                                                                                                                                                                                                                                                                                                                                                                                                                                                                                                                                                                                                                                                                                                                                                                                                                                                                                                                                                                                                                                                                                                                                                                                                                                                                                                                                                                                                                                                                                                                                                        | 带格式的: 字体: 10 磅                                |
|------------|----------------------------------------------------------------------------------------------------------------------------------------------------------------------------------------------------------------------------------------------------------------------------------------------------------------------------------------------------------------------------------------------------------------------------------------------------------------------------------------------------------------------------------------------------------------------------------------------------------------------------------------------------------------------------------------------------------------------------------------------------------------------------------------------------------------------------------------------------------------------------------------------------------------------------------------------------------------------------------------------------------------------------------------------------------------------------------------------------------------------------------------------------------------------------------------------------------------------------------------------------------------------------------------------------------------------------------------------------------------------------------------------------------------------------------------------------------------------------------------------------------------------------------------------------------------------------------------------------------------------------------------------------------------------------------------------------------------------------------------------------------------------------------------------------------------------------------------------------------------------------------------------------------------------------------------------------------------------------------------------------------------------------------------------------------------------------------------------------------------------|------------------------------------------------------|
| 902        |                                                                                                                                                                                                                                                                                                                                                                                                                                                                                                                                                                                                                                                                                                                                                                                                                                                                                                                                                                                                                                                                                                                                                                                                                                                                                                                                                                                                                                                                                                                                                                                                                                                                                                                                                                                                                                                                                                                                                                                                                                                                                                                      | 带格式的: 字体:小五,检查拼
写和语法                       |
| 903        | Table. 1 Dry biomass of leaves, stems and roots for seedlings of Ulmus pumila var.sabolusa exposed to various sand burial treatments                                                                                                                                                                                                                                                                                                                                                                                                                                                                                                                                                                                                                                                                                                                                                                                                                                                                                                                                                                                                                                                                                                                                                                                                                                                                                                                                                                                                                                                                                                                                                                                                                                                                                                                                                                                                                                                                                                                                                                                 | 带格式的: 行距: 1.5 倍行距                             |
| 904        | during a 45-day-growth period. These treatments included sand burial of seedlings to a depth equivalent to 33 (T33), 67 (T67) or 100%                                                                                                                                                                                                                                                                                                                                                                                                                                                                                                                                                                                                                                                                                                                                                                                                                                                                                                                                                                                                                                                                                                                                                                                                                                                                                                                                                                                                                                                                                                                                                                                                                                                                                                                                                                                                                                                                                                                                                                                | 带格式的:字体:小五,非加粗                                       |
| 504        | during a +3-day growin period. These included said buriar of securings to a depin equivalent to 55 (155), 67 (167) or 100/0